



**A comparison of two methods to quantitatively evaluate the effect of**
**below-cloud evaporation on the precipitation isotopic composition in the**
**semi-arid region of the Chinese Loess Plateau**
Meng Xing[1,2*], Weiguo Liu[1,2,3*], Jing Hu[1,2], Zheng Wang[1,2]
1.State Key Laboratory of Loess and Quaternary Geology, Institute of Earth
Environment, Chinese Academy of Sciences, Xi'an, 710061, China
2.CAS Center for Excellence in Quaternary Science and Global Change,Xi'an,
710061, China.
3. University of Chinese Academy of Sciences, Beijing, 100049, China
Corresponding authors:
Meng Xing          email address: xingmeng@ieecas.cn
Weiguo Liu         email address: liuwg@loess.llqg.ac.cn





Abstract:

Below-cloud evaporation effect could heavily alter the isotope composition of the rain water as it travels from the saturated environment in the cloud towards the surface, especially in the arid and semi-arid regions, and accounts for misinterpreting the isotopic signal. To correctly understand the information contained in the precipitation isotopes, the first step is to qualitatively analyze the below-cloud processes that the raindrops have encountered during their falling, and then to quantitatively compute the below-cloud evaporation ratio of raindrops. Here, based on two-year observations of precipitation and water vapor isotopes in Xi'an, we systematically evaluated the variations of precipitation and water vapor isotopes caused by the below-cloud evaporation effect. The precipitation $\delta^{18}O$ and $\delta^2H$ values range from -18.2‰ to 8.8‰ and -131.7‰ to 61.2‰, respectively, while the water vapor $\delta^{18}O_v$ and $\delta^2H_v$ values range from -29.5‰ to -10.1‰ and -214.9‰ to -63.9‰. Our results suggest that the equilibrium method could be successfully used to predict the ground-level water vapor isotopic composition from precipitation isotopes in semi-arid climates, especially for the winter data. Moreover, by using $\Delta d\Delta\delta$-diagram, our data show that evaporation is the main below-cloud process of raindrops, while snowfall samples retain the initial cloud signal because of less isotopic exchange between vapor and solid phases. In terms of meteorological factors, both temperature, relative humidity, and precipitation amount affect the intensity of below-cloud evaporation. In arid and semi-arid regions, the below-cloud evaporation ratio computed by the mass conservation equation would be overestimated relative to the isotopic method, while relative humidity is the most sensitive parameter in computing the remaining fraction of raindrop mass after evaporation. In this study, the mean remaining fractions of raindrop mass calculated by the isotopic method respectively are 69.2%, 74.5%, 85.2%, and 80.8% in spring, summer, autumn, and winter. The raindrops are weakly evaporated in autumn and winter, and heavily evaporated in spring and summer. Based on water vapor and precipitation isotope compositions, we designed a set of effective methods to evaluate the below-cloud evaporation effect, and this will improve our understanding of the information contained in the isotopic signals of precipitation.





## 1 Introduction

The hydrogen and oxygen isotopes of precipitation are one of the greatly important tools to trace the hydrological cycle and climate change (Bowen et al., 2019; Gat, 1996). For the paleoenvironment, the isotopic signals of precipitation recorded in ice cores (Thompson et al., 2000; Yao et al., 1996), tree rings (Liu et al., 2004; Liu et al., 2017b), speleothems (Cai et al., 2010; Tan et al., 2014), and leaf wax of loess-paleosol deposits (Wang et al., 2018b) and lake sediments (Liu et al., 2017a, 2019) could be used to reconstruct the information of temperature, precipitation, and hydrological regimes in geologic history, as it had participated into the formation or growth of these geological archives. For the modern environment, the isotopic ratios of precipitation could be used to quantitatively constraint the water vapor contribution from the end-members of advection (Peng et al., 2011), evaporation (Sun et al., 2020; Wang et al., 2016a), transpiration (Li et al., 2016a; Zhao et al., 2019), and even anthropogenic activities (Fiorella et al., 2018; Gorski et al., 2015; Xing et al., 2020), as precipitation itself is the importantly consisting parts of the water circulation processes. Due to the limitations in sampling and isotopic fractionation theories, however, there remains large uncertainty (i.e., the remaining fraction of below-cloud evaporation, the moisture recycling ratio, water molecules exchange between the droplet and ambient air, etc.) in deciphering the information contained in precipitation by using hydrogen and oxygen isotope ratios (Bowen et al., 2019; Yao et al., 2013).

Chinese Loess Plateau (CLP) is located in the arid and semi-arid areas, where many studies have suggested that the precipitation isotopic composition has been more or less impacted by the below-cloud evaporation and surface moisture recycling effects (Sun et al., 2020; Wan et al., 2018; Zhang and Wang, 2016). Therefore, before we utilize precipitation stable isotopes to reconstruct the climate changes or to trace the water vapor sources, first we need to have a set of reliable evaluation methods to diagnose whether the isotope ratios of precipitation have been distorted by the below-cloud evaporation effect (Graf et al., 2019; Wang et al., 2016b). Then, we need to quantitatively evaluate how much of the raindrops have been evaporated during their falling. Finally, we are able to use the original precipitation isotopes data, which have been calibrated by the below-cloud evaporation effect, to discuss the regional water vapor sources or the global hydrological cycle. At present, however, there are many efforts to do in the first and second steps.

Over the past decades, to determine whether the hydrometeors have been evaporated





during their falling, most studies depend on a second-order isotopic parameter
(Dansgaard, 1964; Jeelani et al., 2018; Li and Garzione, 2017), deuterium excess
(defined as d-excess= $\delta^2H-8\times\delta^{18}O$). This parameter is representative of the kinetic
fractionations, since $^2H^1H^{16}O$ equilibrates faster than $^1H_2^{18}O$ in different phases (Clark
and Fritz, 1997; Dansgaard, 1964). With both the equilibrium and kinetic effects, the
lighter isotopes ($^1H$ and $^{16}O$) of raindrops preferentially equilibrate or diffuse from the
liquid phase to the gas phase during their falling through unsaturated ambient air, while
the non-equilibrium diffusional process would result in a decrease of d-excess in rain
(FISHER, 1991; Merlivat and Jouzel, 1979). Correspondingly, the non-equilibrium
evaporation effect would cause the increase of deuterium excess in the surrounding
water vapor. The slope of the local meteoric water line (LMWL) has also been widely
used as a metric to infer the below-cloud evaporation effect according to the theory of
water isotope equilibrium fractionation (Chakraborty et al., 2016; Putman et al., 2019b;
Wang et al., 2018a), in which the LMWL's slopes approximately equal to 8.0 belonging
to equilibrium fractionation and that is lower than 8.0 pointing to a non-equilibrium
fractionation, such as the re-evaporation of raindrops. Nonetheless, it should be noted
that a change of air masses (Guan et al., 2013), condensation in supersaturation
conditions (Jouzel et al., 2013), and moisture exchange in the cloud and sub-cloud
layer (Graf et al., 2019) also cause largely spatial variation in slopes and d-excess
values (Putman et al., 2019a; Tian et al., 2018). Traditionally, extensive data on the
isotopic content of the condensed phases (e.g., precipitation, snow, ice core, etc.) have
been widely used to study the mechanisms of the atmospheric transport process of
water vapor and the subsequent phase changes in the atmosphere. Inevitably,
important information will lose by using the isotopic composition of the liquid or solid
water samples only. As an improvement, simultaneous observations of water vapor
and precipitation are applied to distinguish these processes and quantify below-cloud
processes. For example, Yu et al. (2015, 2016) used a custom-made sampling device
to collect daily water vapor samples over the Tibetan and Pamir Plateau, and
discussed moisture source impacts on the precipitation isotopes. Using a three-stage
Caltech Active Strand Cloud water Collector (CASCC), Spiegel et al. (2012b, 2012a)
investigated the impact of different processes within clouds, and found that the origin
of the water vapor forming near-surface clouds (fog) is key in determining the temporal
evolution of cloud water isotopes. With the aid of the off-line water vapor sampling
system, Deshpande et al. (2010) analyzed the rain-vapor interaction using stable
isotopes. However, the old water vapor cryogenic trapping technique is time-
consuming (Christner et al., 2018), labor-intensive (Welp et al., 2012), and discrete





(Wen et al., 2016), limiting the further examination of the two-phase system.

In recent years, with the progress in optical laser systems, the relatively portable field-
deployable laser spectroscopic instruments, simultaneously measuring $^1H_2^{16}O$,
$^2H^1H^{16}O$, and $^1H_2^{18}O$ isotopes, allows performing online, autonomous, and long-term
site measurements of the water vapor stable isotope composition (Aemisegger et al.,
2012; Christner et al., 2018). The emergence of this instrument exerts a great impact
on the study of water vapor isotopic composition, leading to a substantially increased
number of observations in near-ground water vapor, while the interpretation of water
vapor isotopic data has the potential to deepen our cognition in water vapor isotopic
variations and fractionation processes during the two-phase transformation (Noone et
al., 2011; Steen-Larsen et al., 2014). Wen et al. (2010) first analyzed the d-excess$_{vap}$
(denotes the d-excess of water vapor) at hourly temporal resolution in Beijing, China,
and systematically discussed the controls on the isotopic exchange between vapor
and condensed phase. Griffis et al. (2016) used multi-year water vapor and
precipitation stable isotope results to evaluate the water vapor contributions to the
planetary boundary layer from evaporation in Minnesota, United States. Laskar et al.
(2014) and Rangarajan et al. (2017) comprehensively investigated the water vapor
sources and raindrop-vapor interaction in Taibei, and developed a box model to explain
the controlling factors for high and low d-excess$_{vap}$ events in this region. Combined
with observations and numerical simulations of stable isotopes in vapor and rain
impacted by cold fronts, Aemisegger et al. (2015) clearly revealed the importance of
below-cloud processes for improving the simulations. An overview of the increasing
number of available water vapor isotope observations can be found in Wei et al. (2019).
As a creative work, Graf et al. (2019) introduced a new interpretive framework to
directly separate the convoluted influences on the stable isotopic composition of vapor
and precipitation according to the theoretical fractionation processes, especially the
influences of equilibration and below-cloud evaporation, which enables us to
disentangle the governing below-cloud processes in the course of a rainfall. Although
Graf's et al. (2019) work gives us a new guideline to more accurately judge the
raindrops experienced below-cloud evaporation effect, their work was only validated
on a cold frontal rain event of a short period, and hence more works need to do for
proving the general applicability of their framework.

In order to get the initial signal of precipitation isotopes, it is necessary to quantitatively
assess the impact of below-cloud evaporation on the stable isotopes. The model





suggested by Stewart (1975) has been widely used to calculate the below-cloud evaporation ratio of raindrops, as the raindrops experienced physical processes have been explicitly described by this isotope-evaporation model (Müller et al., 2017; Sun et al., 2020; Zhao et al., 2019). Based on Stewart's (1975) work, the remaining fraction of raindrop mass ($F_r$) after evaporation could be calculated according to the differences between the stable isotope ratios in collected precipitation near the ground and below the cloud base (See Data and Methods, section 2.3.2, eq 7). We note that some of the studies used the mass conservation model of a falling raindrop to calculate $F_r$ (See Data and Methods, section 2.3.3, eq 8; Kong et al., 2013; Li et al., 2016; Sun et al., 2019; Wang et al., 2016b), and some of the works assumed the $F_r$ is a constant (Müller et al., 2017), but no work has been reported by using ground-based and cloud-based observations of water vapor isotopes to calculate the $F_r$ according to our knowledge. Due to the numerous uncertainty of the parameters in the mass conservation model, such as the factors of terminal velocity, the evaporation intensity, and the diameter of the raindrops, the error propagation will largely raise the deviation of $F_r$ in the model. So far, no work has systematically evaluated the differences of $F_r$ computed by the observed isotope results and the classical mass conservation model.

Here, we have measured the near-ground water vapor isotope composition in Xi'an city (34.23°N, 108.88°E), Shaanxi province, located in the CLP, for 2 years, while collecting 141 precipitation samples (including event-based snowfall samples). The objectives of this study are to: 1. test the applicability of the ΔdΔδ-diagram suggested by Graf et al. (2019) when it is used to diagnose the below-cloud processes for our dataset; 2. compare the differences of raindrops below-cloud evaporation ratio calculated by the observed ground-based water vapor isotope composition and the mass conservation model; 3. understand the role of main meteorological factors, such as temperature, relative humidity (RH), and precipitation amount, on the below-cloud evaporation effect, and the seasonal variations of below-cloud evaporation ratio in CLP. With the advantages of the paired observations of the vapor and precipitation in stable isotopes near the ground level, this study will provide a new set of methods to determine the below-cloud evaporation effect qualitatively. Meanwhile, combined with quantitative calculation, our insight into the below-cloud evaporation effect on the isotopic composition of precipitation in arid and semi-arid areas would be deepened and strengthened.

**2 Data and methods**

**2.1 Sampling site**

As the capital city of Shaanxi province and the largest city in northwest China, Xi'an is located on the Guanzhong Plain on the southern edge of the CLP at an average elevation of 400 m. The city is located in a semi-arid to arid region and is representative of most cities in the north and northwest of China (e.g., Lanzhou and Xining city, Fig. 1). The mean annual precipitation is 573.7mm, and the mean annual evaporation is 426.6mm from 1951 to 2008 year (Wu et al., 2013). The notable below-cloud evaporation effect has been reported by many studies in this area (Sun et al., 2020; Wan et al., 2018; Zhu et al., 2016).

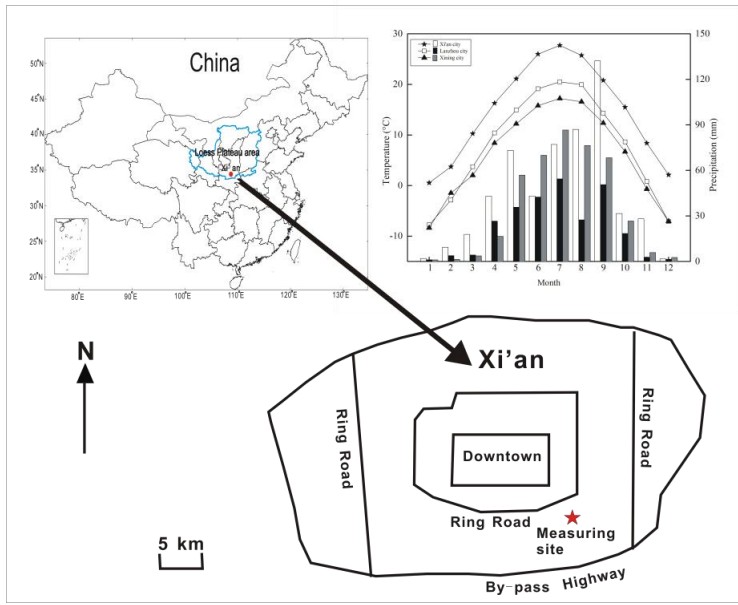

Figure 1 Average monthly variations of temperature and precipitation in Xi'an, Lanzhou, and Xining during 2010-2015. Location of the sampling site in the Yanta Zone, 9 km SE of downtown Xi'an. Water vapor samples are taken on the seventh floor of a twelve-story building, about 30 m above ground level. Precipitation samples are collected on the top floor, 1 m above ground level.

The water vapor in-situ measurement is located in a residential area, approximately 10 km southeast to downtown of Xi'an city (Fig. 1). The atmospheric water vapor isotopic composition was observed from 1 January 2016 to 31 December 2017 on the seventh floor of the Institute of Earth and Environment, Chinese Academy of Sciences, about 30 m above ground level. The rainfall or snowfall collector was placed on the rooftop of the buildings (1 m above the floor of the roof).





**2.2 Sampling and isotopic measurement**

Rainfall and snowfall samples were collected manually from the beginning of each precipitation event using a polyethylene collector (700 mm × 450 mm × 170 mm) and the volume was measured using a graduated flask. Before being used, the collector was cleaned with soap and water, rinsed with deionized water, and then dried. When the precipitation events end, the collector was quickly taken back to minimize water evaporation. Rainfall samples were immediately poured into a 100 ml polyethylene bottle. The snowfall samples were melted at room temperature in a closed plastic bag after collection, and then immediately poured into a 100 ml polyethylene bottle. After collection, samples were filtered through 0.40–μm polycarbonate membranes. About a 2 ml of each filtrate was transferred into a sample vial, and stored at – 4°C until being measured. Of the 141 samples, during the two-year sampling campaigns, we collected 130 rainfall and 11 snowfall samples (Table S1).

In all cases, the data are reported in the standard delta notation (δ), i.e., the per mil (‰) deviation from Vienna Standard Mean Ocean Water according to, δ= $(R_{sample}/R_{reference}-1) \times 1000$, where R is the isotope ratio of the heavy and light isotope (e.g., $^{18}O/^{16}O$) in the sample and the reference.

The precipitation samples were measured by Picarro L2130-i wavelength-scanned cavity ring-down spectrometer at a high-precision model. Every isotopic standard or sample was injected sequentially 8 times using a 5 μL syringe, and then the arithmetic average of the last 3 injections was accepted as the final result. The precision is better than 0.2‰ and 1.0‰ for $\delta^{18}O$ and $\delta^2H$, respectively. All the samples were calibrated by three laboratory standards, while the $\delta^{18}O$ and $\delta^2H$ true values of the three laboratory standards (Laboratory Standard-1 (LS-1): $\delta^{18}O$ =+0.3‰, $\delta^2H$ =–0.4‰; Laboratory Standard-2 (LS-2): $\delta^{18}O$ =–8.8‰, $\delta^2H$ =–64.8‰; Laboratory Standard-3 (LS-3): $\delta^{18}O$ =–24.5‰, $\delta^2H$ =–189.1‰) are calibrated to the scale of two international standard material VSMOW-GISP, with a precision of ±0.2‰ and ±1.0‰, for $\delta^{18}O$ and $\delta^2H$, respectively.

Atmospheric water vapor $\delta^{18}O_v$ and $\delta^2H_v$ were also measured by Picarro L2130-i (serial number HIDS 2104), but at a liquid-vapor dual model. The inlet of the gas-phase instrument is connected to the vapor source through an external solenoid valve when measuring vapor samples. This valve can switch the input of the instrument from the vapor sample to dry gas. The instrument is connected to dry gas prior to being





connected to the evaporator for measuring liquid water standards so that any traces of
the water vapor sample are removed from the measurement cell. The standards are
injected into the evaporator and measured by a CTC Analytics autosampler, PAL HTC-
xt (Leap Technologies, Carrboro, NC, USA). The atmospheric water vapor is pumped
through a 2m stainless-steel tube (1/8 inch) using a diaphragm pump at the speed of
4 L min$^{-1}$ and detected by the laser spectrometer. The outside length of the stainless-
steel tube is about 0.5m, and the inside length is about 1.5m. We covered the stainless-
steel tube with a heating tape maintained at 60°C to prevent water vapor from
condensing in the stainless-steel tube. The air intake was protected with a shield to
prevent rainwater from entering the sample line and direct sunlight.

The raw water vapor $\delta^{18}O_v$ and $\delta^2H_v$ data were obtained approximately at 1 Hz and
then block-averaged into 24 h intervals. As the main usage of this instrument is to
measure the liquid water samples in our laboratory, it is used to monitor the water
vapor isotopes in its spare time. Thus, the data gaps represent the instrument is in
liquid samples measuring status or maintenance. The daily average of water vapor
isotopic composition is from 8:00 - 20:00 UTC (0:00 – 24:00 for local time).  The
average intra-day variability of water vapor isotopic composition is less than 1.2‰ for
$\delta^{18}O$ and 8.4‰ for $\delta^2H$ for two-year data, respectively, and on the precipitation day is
1.4‰ for $\delta^{18}O$ and 10.5‰ for $\delta^2H$, respectively.

The hourly meteorological data, such as temperature and RH in Xi'an, are reported by
the China meteorological administration, and can be downloaded from the website of
http://www.weather.com.cn/. The meteorological station is about 10 km to the north of
our sampling site.

Here, we need to note the different sampling frequencies between Graf's et al., (2019)
and our study. To explicitly capture the below-cloud processes of the droplet, Graf's et
al., (2019) study used the intra-event samples, which clearly record the equilibration,
evaporation, and cloud signal on the ΔdΔδ-diagram. Particularly, the below-cloud
evaporation effect does not accompany the entire rainfall process, for example, in the
pre-frontal phase, the rain intensity and relative humidity are lower than in the post-
frontal period, which causes the raindrops to be more strongly affected by below-cloud
processes (Graf et al., 2019). In this study, we aim to quantitatively evaluate the
evaporated degree of droplets in a single rain event, and compare the results of two
different methods for calculating the remaining fraction of raindrop mass after





evaporation. The per-event isotopic composition of precipitation is an integrated,
mass-weighted average of the composition of all drops contained in a sample (Graf et
al., 2019). The processes that act on a single drop are thus directly relevant for bulk
precipitation. In the per-event sample, the offset between the precipitation equilibrated
isotope ratios and the simultaneously observed isotope ratios of surface vapor can aid
in inferring the below-cloud processes (Conroy et al., 2016). Hence, we chose the per-
event samples as our study objects. Note that, the per-event samples whose isotopic
results do not project on the fourth quadrant of ΔdΔδ-diagram, do not indicate the
absence of below-cloud evaporation. It rather is an indication that the equilibration or
the cloud isotopic signals dominate the mass-weighted isotopic composition of all
drops.

### 2.3 Water vapor isotopic data calibration

Due to the isotopic measurements of the cavity ringdown spectrometer with water
vapor concentration effect as outlined by some studies (e.g., Bastrikov et al., 2014;
Benetti et al., 2014; Steen-Larsen et al., 2013), it is important to determine the
humidity-isotope calibration response function. Because we did not have the
Standards Delivery Module (Picarro) system or equivalent, the humidity calibration is
based on data obtained from discrete injections of three known liquid standards with a
PAL autosampler and the Picarro vaporizer unit (Benetti et al., 2014; Noone et al.,
2013). The analyzer is programmed to perform a self-calibration after every 24 hours
of ambient air measurement using an autosampler to inject liquid standards for
producing different humidity. Injections were arranged at humidity levels near 3000,
5000, 8000, 10000, 15000, 20000, 25000, and 30000 ppm. Each reference sample is
measured continuously for 8 times at one humidity level, and the last 3 times results
were used to calculate the average to be recognized as the δ-value at the measured
humidity. The humidity correction is the difference between the δ-value at the
measurement humidity and the δ-value at a reference value taken as humidity = 20000
ppm. The best fit was reached with an exponential function for $\delta^{18}O_v$ and a linear
function for $\delta^2H_v$ (Fig. S1a and S1b). The isotopic measurements of ambient air $\delta^{18}O_v$
samples were corrected for humidity effects using:
$\delta^{18}O_{\text{humidity calibration}} = \delta^{18}O_{\text{measured}} - (-4.91 \times e^{(-3.51 \times \text{Measured humidity})})$      (eq 1)
and for ambient air $\delta^2H_v$ humidity correction using:
$\delta^2H_{\text{humidity calibration}} = \delta^2H_{\text{measured}} - (0.0001 \times \text{Measured humidity} - 1.86)$      (eq 2)





where $\delta_{\text{humidity calibration}}$ is the calibrated data for water vapor stable isotope; $\delta_{\text{measured}}$ is
the raw, measured data before calibration; and measured humidity is the
corresponding humidity at the time of measurement.
To calibrate the measured water vapor isotopic composition to the VSMOW-GISP
scale, three known-value laboratory standards have been used in the conversion,
while these standards were measured in 24 h intervals to correct for instrument drifts.
The detailed post-calibration procedure is given in Xing et al. (2020). The 1σ estimated
total uncertainties are from 2.1 to 12.4 ‰ for $\delta^2H_v$, 0.4 to 1.7 ‰ for $\delta^{18}O_v$, and 3.8 to
18.4 ‰ for d-excess$_v$ over the range of humidity from 30000 to 3000 ppm on a 10-
minutes average through the approach using a Monte Carlo method.

**2.4 The representative of the data**
During the two-year study, we collected the precipitation samples for each event.
Precipitation samples are generally collected from the beginning of the rainfall to the
end. If the rainfall event exceeds 24 hours, we replace a sample collector at 8 am as
a new precipitation sample. For the observation of water vapor isotopic composition, it
has been done in the instrument's spare time, that is when the instrument is not on
liquid water samples testing mission or maintaining status. In 2 years, a total of 514
days of water vapor isotopic composition were carried out, of which 100 precipitation
samples have corresponding daily average water vapor isotopic results. In this study,
the precipitation events mainly occurred in summer and autumn, and less in winter and
spring. The rainfall amount accounted for more than 70% of the annual rainfall in
summer and autumn (Fig. S2). This is consistent with the multi-year average
precipitation distribution in Xi'an (Fig. 1). Therefore, the samples we collected are
representative of the precipitation characteristics of this region.

**2.5 Analytical methods**
**2.5.1 ΔdΔδ-diagram**
As the raindrop is falling from the cloud base to the ground, it continuously exchanges
with the surrounding vapor, but may lead to net loss as evaporation. However, this
process is very hard to be quantified by observation. Using stable water isotopes, Graf
et al. (2019) introduced a ΔdΔδ-diagram to diagnose below-cloud processes and their
effects on the isotopic composition of vapor and rain since equilibration and
evaporation are two various below-cloud processes and lead to different directions in
the two-dimensional phase space of the ΔdΔδ-diagram. Here, the differences of
isotopic composition of equilibrium vapor ($\delta^{18}O_{pv-eq}$, d-excess$_{pv-eq}$) from precipitation


samples relative to the observed ground-based water vapor ($\delta^{18}O_{gr\text{-}v}$, d-excess$_{gr\text{-}v}$) can
be expressed as:

$$\Delta\delta=\delta_{pv\text{-}eq} - \delta_{gr\text{-}v} \qquad\qquad (eq3)$$


$$\Delta d\text{-}excess_v = d\text{-}excess_{pv\text{-}eq} - d\text{-}excess_{gr\text{-}v} \qquad (eq4)$$


where $\delta_{pv\text{-}eq}$ and $\delta_{gr\text{-}v}$ are the $\delta^2H$ ($\delta^{18}O$) of water vapor below the cloud base and near
the ground, respectively, and d-excess$_{pv\text{-}eq}$ and d-excess$_{pv\text{-}eq}$ are d-excess values of
water vapor below the cloud base and near the ground, respectively.

To calculate the water vapor isotopic composition below the cloud base, we
hypothesize the constant exchange of water molecules between the liquid and the
vapor phases during the falling of raindrops, and the isotopic compositions reach
towards equilibrium in the two phases during the processes. In the equilibrium state,
the isotopic fractionation between the liquid and vapor phases follows a temperature-
dependent factor:

$$\alpha \;\; = \frac{R_{pv\text{-}eq}}{R_p} \qquad\qquad (eq5)$$


where $R_{pv\text{-}eq}$ is the water vapor isotope ratio between heavy and light isotopes ($^2H/^1H$
and $^{18}O/^{16}O$), $R_p$ is the isotope ratio in precipitation, and $\alpha$ is a temperature-dependent
equilibrium fractionation factor. Here, when the temperature is greater than 0 °C, we
use the equation of  Horita and Wesolowski (1994) to calculate $^2\alpha$ and $^{18}\alpha$, when the
temperature is below 0 °C, the equilibrium fractionation factor proposed by Ellehoj et
al. (2013) is used.

The above equation can be converted into $\delta$-notation as:

$$\delta_{pv\text{-}eq} = \frac{1}{\alpha} \left(\delta_p + 1000\right) - 1000 \qquad\qquad (eq6)$$


where $\delta_p$ is the isotope ratio in precipitation.

### 2.5.2 Below-cloud evaporation calculated by isotope

Stewart (1975) suggested the falling raindrop isotopic fractionation of evaporation
could be calculated according to the fraction of raindrop mass remained after
evaporation:

$$\Delta\delta = \delta_p - \delta_{zp\text{-}eq} = \left(1 - \frac{\gamma}{\alpha}\right)\left(F_{iso}^{\beta} - 1\right) \qquad\qquad (eq7)$$


where $\delta_p$ and $\delta_{zp\text{-}eq}$ are precipitation isotope ratio near the ground and below the cloud
base, respectively; $F_{iso}$ is the remaining fraction of raindrop mass after evaporation



(hereafter, the remaining fraction of raindrop mass calculated by this method is
denoted as $F_{iso}$); α is equilibrium fractionation factor for hydrogen and oxygen isotopes;
the parameters of γ and β is defined by Stewart (1975). For the detailed calculation
processes, please refer to the supplemental material (Appendix A), or Wang et al.
(2016b), Salamalikis (2016), Graf et al. (2019), and Sun et al. (2020).
**2.5.3 Below-cloud evaporation calculated by mass conservation model**
Before the advent of the laser-based spectrometer, the water vapor isotopic
composition measurement is labor-intensive and time-consuming, generally using the
custom-made cold trap to collect. Normally, its observation is not a routine option.
Therefore, to correct the measured δ/d-excess in precipitation (eq7) for the effect of
below-cloud evaporation (Kong et al., 2013; Li et al., 2016a; Zhao et al., 2019) or study
the differences and controls on δ/d-excess in precipitation caused by the below-cloud
evaporation (Wang et al., 2016b), the Stewart (1975) model have been widely used.
In the model, the parameter of the remaining fraction of the water-drop mass is variable
and decisive (eq 8), and can be calculated by the law of conservation of mass
(hereafter, the remaining fraction of raindrop mass calculated by this method is
denoted as $F_{raindrop}$),
$$F_{raindrop} = \frac{m_{end}}{m_{end} + m_{ev}} \qquad (eq8)$$
Here, the mass of the reaching ground raindrop after evaporation is $m_{end}$ and the
evaporated raindrop mass is $m_{ev}$. The parameter of $m_{ev}$ is composed of the evaporation
rate and fall time of the drop. Kinzer and Gunn (1951) and Best (1950a) have
parameterized these two variables, respectively. For the detailed calculation
processes, please refer to the supplemental material (Appendix B), or Wang et al.
(2016b), Sun et al. (2020), Kong et al. (2013), and Salamalikis (2016).
Due to the  intrinsic limitations of Stewart's model, which is a mixing model between
the starting and the final isotopic composition, it assumes a homogenous sub-cloud
layer in terms of temperature and humidity (Salamalikis et al., 2016). Here, considering
the location of our study site, we reasonably assume that precipitation forms close to
the average cloud base at about the 850 hPa (~1500 m) isobaric level (Kong et al.,
2013; Li et al., 2016a; Salamalikis et al., 2016). The cloud base temperature and RH
were obtained from the moist adiabatic ascent of an air parcel from the surface with
initially measured temperature and RH (Appendix A). It is worth noting that the results
may be affected by errors originating from assumed cloud base heights and calculated
vertical profiles of temperature and RH. In the future, the pieces of evidence from direct





measurement of cloud base heights, temperature, and RH from radiosondes or aircraft
will validate the assumed below-cloud profiles presented here.

Now, with the emergence of a laser-based spectrometer, high-precision, high-
resolution water vapor isotopic composition measurement is becoming easier, and
thus the evaluation of below-cloud evaporation is becoming more direct. However, little
work has systematically evaluated the differences of $F_r$ computed by different methods.
Here, we respectively used the isotope and mass conservation methods to calculate
the $F_r$, and compared their differences.

**2.5.4 Statistical Analysis**
To compare the difference of the below-cloud evaporation calculated by the two
methods, the independent t-test was performed on SPSS 13.0 (SPSS Inc., Chicago,
US). A significant statistical difference was set at $p < 0.05$.

**3 Results and discussion**
**3.1 Relationship between water vapor and precipitation isotopic ratios**
Influenced by the below-cloud evaporation, the slope of the local meteoric water line
(LMWL) would be lower than 8, the precipitation isotopic composition become more
positive, the d-excess of precipitation would be less than 10, and the equilibrium
calculated water vapor isotopic composition would be more positive than the observed
one. As shown in Figure 2, the LMWL is: $\delta^2H_p=7.0\times\delta^{18}O_p+3.0$ based on event
precipitation isotopic composition, and the local water vapor line (LWVL) is:
$\delta^2H_v=7.6\times\delta^{18}O_v+10.0$ based on daily water vapor isotopic composition. Both the slope
and intercept of LMWL are lower than the Global Meteoric Water Line (GMWL) which
are 8.0 and 10.0 (Dansgaard, 1964; Gat, 1996), respectively, indicating the potentially
significant effect of below-cloud evaporation on precipitation (Froehlich et al., 2008). In
general, the slopes of the meteoric water lines are indicative of kinetic processes
superimposed on the equilibrium fractionation, and the little lower slope of LWVL
(slope=7.6) than the expected equilibrium fractionation (slope=8.0) may also relate to
the increasing influence of kinetic processes (Rangarajan et al., 2017).






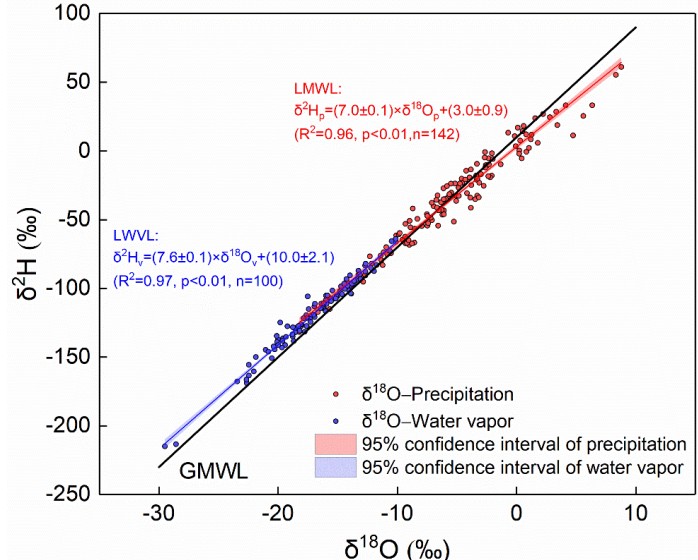

Figure 2 Local meteoric water line (LMWL) and Local water vapor line (LMVL) in Xi'an.

Besides, we note that the water vapor and precipitation isotopic composition basically
distribute in different ranges, and the water vapor isotopic composition is generally
more negative than the precipitation isotopic composition (Fig. 2). According to the
classical isotopic fractionation theory, the heavier isotopes preferentially condense into
the liquid phase resulting in the precipitation isotopic ratios more positive than the
corresponding water vapor during the precipitation process (Dansgaard, 1964). Hence,
the perfect distribution characteristics of water vapor and precipitation on the $\delta^{18}O$-$\delta^2H$
plot would make us suppose that the precipitation and water vapor isotopic
composition are in or close to equilibrium in this study site. To validate our assumption,
we plot the relationship between the per-event precipitation and the corresponding
day's water vapor isotopic compositions in Figure 3a, as expected they show a
significant positive correlation ($R^2$=0.66, p<0.01). The water vapor isotopic composition
can explain above 60% of the variation of precipitation isotopic composition. Further,
we use the measured precipitation isotopic composition to deduce the water vapor
isotopic composition at the cloud base (1500m) according to the liquid-vapor
equilibrium isotope fractionation, and compare it with the observed near-ground water
vapor isotopic composition. As expected, the scatterplot of the observed $\delta^{18}O_v$ against
the deduced $\delta^{18}O_{v-eq(1500m)}$ also presents a significantly positive relationship, and the
correlation coefficient increases by 4%  (Fig. 3b).




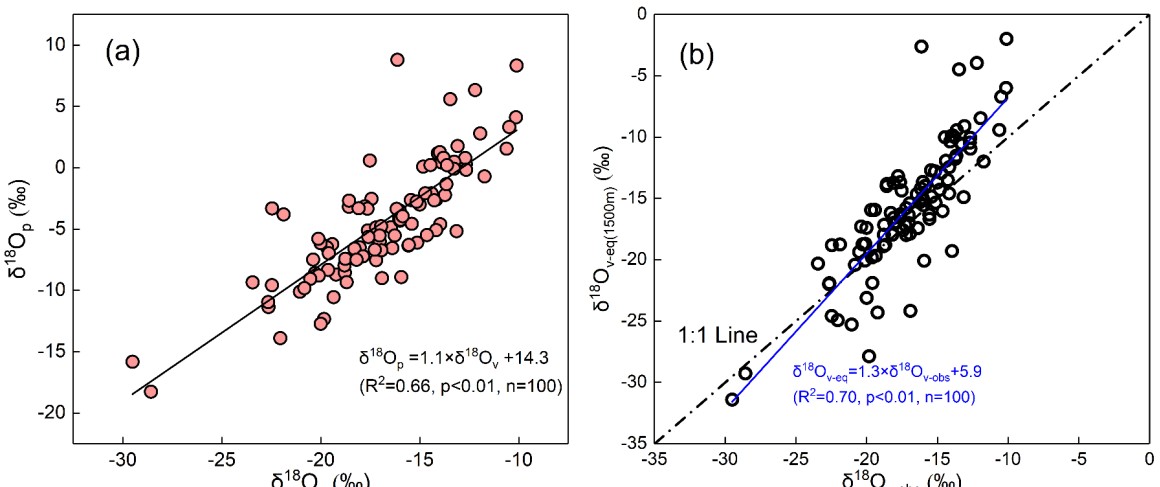

Figure 3 Relationship between $\delta^{18}O_p$ of precipitation and $\delta^{18}O_v$ of water vapor in Xian (a); and
relationship between the equilibrium computed 1500m $\delta^{18}O_v$ based on the precipitation isotopic
composition and the near ground observed $\delta^{18}O_v$ (b). The dash-dot line in (b) stands for the 1:1
line, and the blue line represents the regression line of the data.

The reasonable agreement of $\delta^{18}O$ between observed water vapor and equilibrium
prediction has been reported by Jacob and Sonntag (1991), Welp et al. (2008), and
Wen et al. (2010), however, they postulated the different relationship underlying the
$\delta^{18}O_v$ and $\delta^{18}O_{pv-eq}$. Jacob and Sonntag (1991) suggested that the water vapor isotopic
composition is possible to be deduced from the corresponding precipitation isotopic
composition, but Wen et al. (2010) speculated that the equilibrium method cannot
accurately predict the ground-level water vapor isotopic composition in arid and
semiarid climates because of two monthly equilibrated precipitation values deviating
the observed water vapor values. Here, with two-year continuous observation, the
mean difference between the $\delta^{18}O_{v-obs}$ and $\delta^{18}O_{v-eq(1500m)}$ is $-1.1‰$ for $\delta^{18}O$, -8.1‰ for
$\delta^2H$, and 0.7‰ for d-excess, and our results indicate that it is possible to derive the
isotope composition of atmospheric water vapor based on that of the precipitation in
the semi-arid area. It is worth noting that we do not propose to extract the water vapor
isotope time series from precipitation data. Because, in dry regions of the world,
precipitation events are rare so deriving vapor isotopes from precipitation can be very
misleading. No data is available for the sometimes long dry spells without precipitation.
These periods are likely to exhibit very special vapor isotope signals about which no
information can be gained from precipitation data.





In addition, we also noted that the equilibrium calculated $\delta^{18}O_{v\text{-}eq(1500m)}$ is relatively more
positive than the $\delta^{18}O_{v\text{-}obs}$ (Fig. 3b). In theory, the water vapor isotopic composition
decreases with altitude (Deshpande et al., 2010; Salmon et al., 2019). However, due
to the CLP belonging to the semi-arid area, the raindrops are likely to experience
evaporation in the unsaturated. Therefore, the positively equilibrated $\delta^{18}O_{v\text{-}eq(1500m)}$ is
caused by the kinetic fractionation in low relative humidity, and this also makes the
$\delta^{18}O_{v\text{-}eq}$-$\delta^{18}O_{v\text{-}obs}$ points deviate from the 1:1 line.

**3.2 Below-cloud processes indicated by ΔdΔδ-diagram**
Traditionally, to qualitatively assess the below-cloud evaporation of raindrops, the
value of d-excess$_p$ is a benchmark, as the isotopically kinetic fractionation will cause
d-excess$_p$ to deviate from 0‰, which is a theoretical value under vapor-liquid
equilibrium fractionation (Gat, 1996). The global mean value of 10‰ for the d-excess
in precipitation indicates that evaporation is in general a non-equilibrium process.
Normally, below-cloud evaporation will move d-excess$_p$ below 10‰, and in comparison,
mixing with the recycled water vapor from surface evaporation and plant transpiration
will bring d-excess$_p$ above 10‰ (Craig, 1961; Dansgaard, 1964). Kinetic (non-
equilibrium fractionation) is due to the differences in diffusivities of the individual water
molecules. Therefore, during the moisture transportation, the water vapor d-excess
may be modified, and this enhances the uncertainty to gauge the below-cloud
evaporation process by solely using d-excess$_p$. In contrast, the ΔdΔδ-diagram
introduced by Graf et al. (2019) provides richer information on the below-cloud
processes.

By projecting our data on the ΔdΔδ plot, the evaporation, equilibration, and non-
exchange (e.g., a snowfall event, or a transition from rain to snow with a stronger cloud
signal) processes could be clearly differentiated (Fig. 4). It is apparent from Fig. 4, that
most of the precipitation samples are located in the fourth quadrant, indicating that
evaporation is the dominant below-cloud process. A small part of the samples is
distributed in the first and second quadrant, and their Δδ are close to 0‰ while Δd are
a little higher than 0‰. This cluster of samples implies that the below-cloud evaporation
and cloud-based isotopic fractionation tend to achieve a complete equilibrium state.
Interestingly, in our samples, most of the snow samples seize the third quadrant, which
is suggestive of below-cloud evaporation with less impact on them, and their initial
signal after cloud-based equilibrium fractionation is well retained. According to results
from numerical simulations and in-situ observations, Graf et al. (2019) summarized
that raindrop size and precipitation intensity appear to be the important driving factors
of the below-cloud processes, because raindrops with large diameter and high
intensity will reduce their residence time in the atmospheric column, and lower the
evaporation possibility during its way down toward the ground surface. However, as
for snowfall events, it seems unreasonable to explain the strongly negative Δδ from
through the drop size and rain rate (Fig. 4). It is well known that snowfall events
generally happen in low-temperature conditions, and correspond to weak evaporation,
due to the lower diffusion speed of the ice phase to vapor as compared to liquid to
vapor. Hence, rain/snow formed under such circumstances, its isotopic signals will not
be largely changed by the environmental factors during its falling, which leads the Δδ
to be more negative with the decrease of temperature, such as the phenomenon
observed in Graf's et al. (2019) study during the post-frontal periods. Our results
suggest that in addition drop size and rain rate, precipitation type is also an essential
factor that needs to be fully considered in the below-cloud processes.

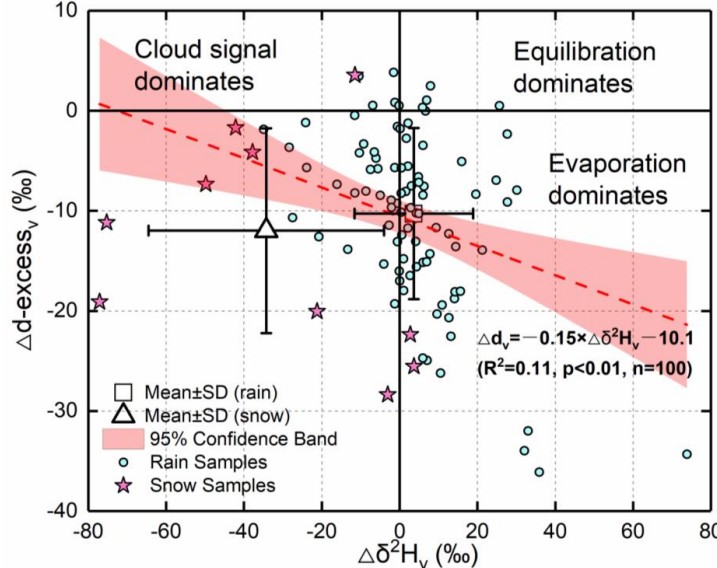

Figure 4 The projection of our data on Graf et al. (2019) suggested ΔdΔδ-diagram. The solid
lines stand for Δd-excess$_v$ and Δδ$^2$H$_v$ of 0‰. The dashed line corresponds to the linear fit
through the samples with the 95% confidence band in red shading.

Meteorological factors, such as precipitation amount, temperature, and RH, are the
main factors affecting below-cloud evaporation (Li et al., 2016b; Peng et al., 2007),
and have been well studied by combined with precipitation d-excess$_p$ (Ma et al., 2014;
Wang et al., 2016b). In order to further analyze the below-cloud processes, we add the
meteorological and isotopic information on the ΔdΔδ-diagram (Fig. 5). Generally, with





regard to high $\Delta^2H_v$ samples, the corresponding meteorological condition is high
temperature, low precipitation amount, and low RH (Fig. 5a-c). In contrast, under a
condition of low air temperature, high RH, and large precipitation amount, the $\Delta^2H_v$ of
samples are relatively more negative (Fig. 5a-c). As below-cloud processes are
controlled by multi-variable factors, it is hard to only use single physical variable to
explain the below-cloud evaporation (Ma et al., 2014; Wang et al., 2016b). For example,
under the highest temperature condition (two most red dots in Fig. 5a), the below-cloud
evaporation effect should be higher, and cause $\Delta^2H_v$ to be more positive and $\Delta$d-
excess$_v$ to be more negative. However, under such circumstances, both the $\Delta^2H_v$ and
$\Delta$d-excess$_v$ of the two samples are close to 0‰. By considering the precipitation
amount, the two samples collected under the highest temperature condition are
associated with a relatively larger precipitation amount which will temper the intensity of
below-cloud evaporation. In addition, higher temperature corresponds to higher
saturation vapor pressure, and a larger number of water molecules present in the
atmosphere, which may enable substantial, rapid equilibration of water molecules
between raindrops and ambient vapor during fall. Similarly, the samples with lower
precipitation amount are associated with high RH, and cause the $\Delta^2H$ distributed
around 0‰. For the snow samples, the data  with positive $\Delta^2H$ is related to the very
low RH (Fig. 5c).

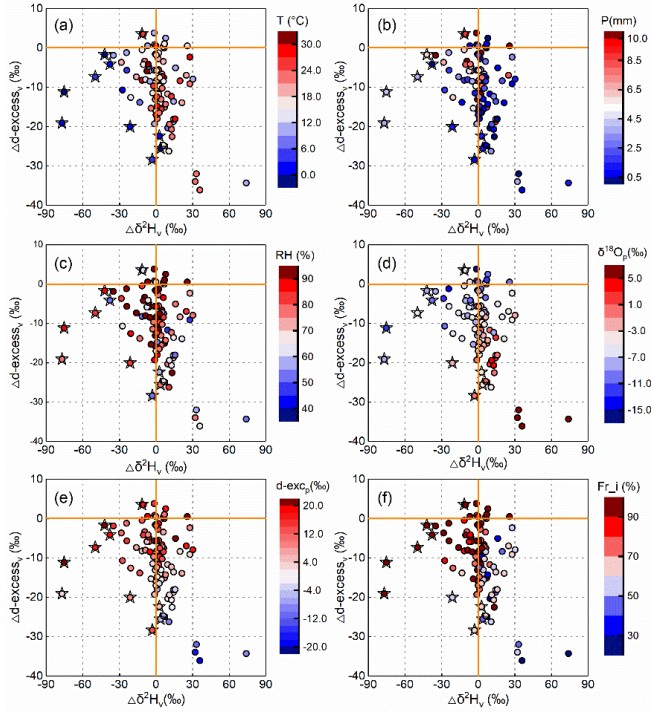





Figure 5 ΔdΔδ-diagram for the precipitation samples with meteorological factors and
precipitation isotopic information. Temperature (a); Precipitation amount (b); Relative humidity
(c); $\delta^{18}O_p$ of precipitation (d); d-excess$_p$ of precipitation (e); Remaining fraction of evaporation
(f). The dots with a star represent the snow samples.
In contrast to meteorological factors, the pattern of precipitation isotopic composition
distribution on the ΔdΔδ-diagram is more clear. Under the high below-cloud
evaporation condition, the $\delta^{18}O_p$ is more positive and d-excess$_p$ is relatively negative
(Fig. 5d and 5e). Correspondingly, the differences between equilibrated $\delta^2H_{eq-v}$, d-
excess$_{eq-v}$ and observed $\delta^2H_{gr-v}$, d-excess$_{gr-v}$ are larger. Conversely, under low below-
cloud evaporation conditions, mainly corresponding to the most snow samples, we
could see the lowest $\delta^{18}O_p$ and highest d-excess$_p$ samples, respectively (Fig. 5d and
5e). Moreover, the $\Delta^2H_v$ is lower than 0‰ and Δd-excess$_v$ is placed around 0‰.
Basically, the ΔdΔδ-diagram follows not only the traditional explanation that Δd<0‰
and Δδ>0‰ indicate the below-cloud evaporation process but also provides more
information on the falling raindrops, such as Δd<0‰ and Δδ<0‰ indicating the cloud
signals, and Δd and Δδ close to 0‰ indicating equilibrium conditions.
The slope of the regression line of Δd/Δδ is -0.15 in our study (Fig. 4), which is half of
the slope shown by Graf's et al. (2019). However, the slope of Graf's et al. (2019) is
based on intra-event samples, while ours is on per-event samples, hence the two
slopes cannot compare with each other directly. To advance the understanding of the
slope, the controlling factors have been analyzed.
According to the sensitivity test by Graf et al. (2019), RH has a considerable impact on
the slope of Δd/Δδ. Low RH is coupled with more negative slopes, while the slopes of
Δd/Δδ under high RH conditions are less negative or even positive (Fig. 5c). In addition,
the temperature has a similar impact on the slopes of Δd/Δδ as the RH (Fig. 5a). This
indicates that the negative slopes of Δd/Δδ correspond to a warm and dry environment.
Besides, the slopes of Δd/Δδ may relate to the precipitation types. When the samples
are separated into rainfall and snowfall, the rainfall slope is -0.28 and the snowfall
slope is only -0.12. Although the time scales are different in the two studies,
interestingly, the slopes of rainfall are more close to each other. The slope of -0.3 could
represent a general characteristic of rainfall for continental mid-latitude cold front
passages (Graf et al., 2019), while the slope of snow samples is less negative in our
study (Fig. S3). Certainly, to explore the relationship between the slope of Δd/Δδ and
the climatic characteristics and precipitation types, more validation works need to do





in future studies.

### 3.3 Comparing and analyzing the differences between $F_{iso}$ and $F_{raindrop}$

### 3.3.1 The differences and reasons

In 1975, Stewart (1975) presented a set of empirical models, which is still widely used,
to evaluate the below-cloud evaporation rate of the falling raindrop. However, Limited
by measuring the cloud-based isotopic composition of the raindrop, many studies turn
to use mass conservation model to evaluate the evaporation rate of the raindrop during
its falling (Kong et al., 2013; Li et al., 2016a; Sun et al., 2020; Wang et al., 2016b).
Here, for comparing their differences, we used the isotope method and mass
conservation model to calculate the $F_r$ after the below-cloud evaporation, respectively.
For both methods, we only considered evaporative exchange in below-unsaturation
cases. However, it is not true for the isotopic method, because both the evaporation
and equilibration have effects on the $\Delta\delta$ of the droplet during its falling from the cloud
base. Here, we assumed that the below-cloud precipitation and surrounding ambient
water vapor are in fully isotopic equilibrium, and $F_{iso}$ only accounts for the evaporation
effect on $\Delta\delta$. Therefore, the $F_{iso}$ results may underestimate the remaining fraction of
evaporation. To get the accurate $F_{iso}$ results, more works need to do to partition the
equilibration process from the evaporation in the future.

As shown in Fig. 6a, the computed mean of reaming fraction is 76.3% by the isotopic
method ($F_{iso}$), and 65.6% by the mass conservation model ($F_{raindrop}$) based on two-year
statistical results. The $F_{raindrop}$ is statistically lower than the $F_{iso}$ depending on the
independent t-test (F=1.49, p<0.01). In addition, the $F_{iso}$ and $F_{raindrop}$ show an obvious
difference, and that is the $F_{iso}$ and $F_{raindrop}$ seriously deviating from 1:1 line, when the
$F_{iso}$ equals to 60%~80% (Fig. 6b). On a seasonal scale, the difference between $F_{iso}$ and
$F_{raindrop}$ in spring, summer, autumn, winter is 13.7%, 12.8%, 6.0%, and 25.0%,
respectively, which is the largest in winter, and the lowest in autumn (Fig. 7).

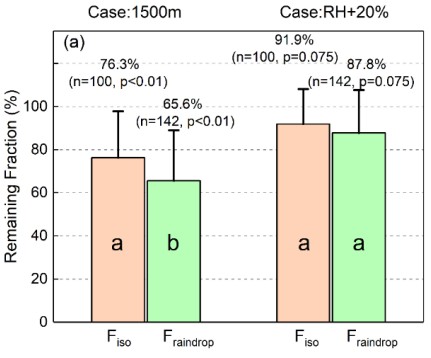
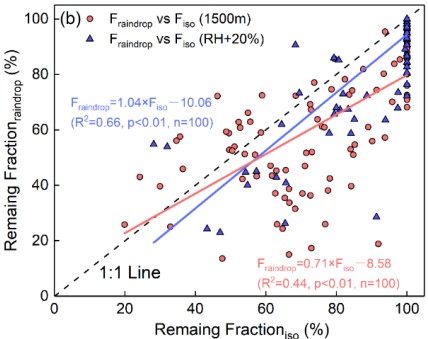



Figure 6 The comparison between the remaining fraction calculated by two methods. In (a), the
taupe bars and the abbreviation of $F_{iso}$ represent the remaining fraction calculated by the
isotopic method, and the green bars and the abbreviation of $F_{raindrop}$ represent the remaining
fraction calculated by the mass conservation method. Case 1500m denotes that the raindrops
evaporation calculation is based on the assumption of the cloud base at 1500m, and the
raindrops are formed at that altitude. Case RH+20% denotes that based on the condition of
case 1500m, we calculated the remaining fraction by increasing the ground observed RH by
20%. n represents the number of samples used in statistics. The a and b in the bars denote the
results of the independent t-test. In (b), the red dots represent the computed results of the
remaining fraction under case 1500m condition, and the blue triangles represent the computed
results under case RH+20% condition. The dash line is the 1:1 line.

To further explore the reason for the large differences by employing different methods,
we performed the correlation analyses between meteorological factors and the
remaining fraction of evaporation (Fig. S4). These analyses reveal that the most
important impact factor both on $F_{iso}$ and $F_{raindrop}$ is RH (Fig. S4b). Although precipitation
amounts have influences on $F_{iso}$ and $F_{raindrop}$ as well, their relationships are non-linear,
and its effect on $F_{iso}$ is rather weak ($R^2$=0.16, Fig. S4c). For temperature, no clear
correlation was found. Wang et al. (2016b) explicitly pointed that among the
parameters of temperature, precipitation amount, RH, and raindrop diameter, RH
generally plays a decisive role on the obtained Δd-excess, which is positively
correlated with the remaining fraction of raindrop.

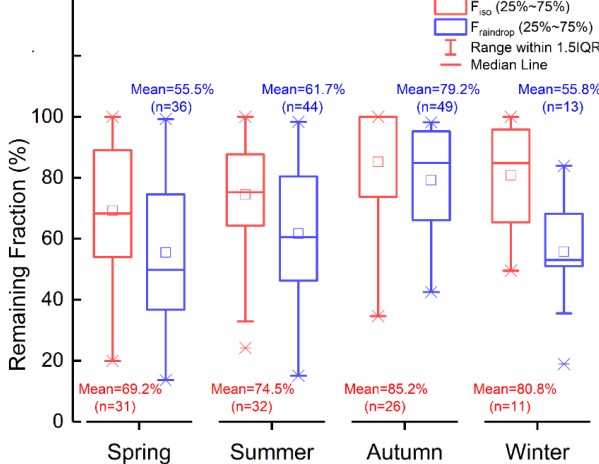

Figure 7 Comparison between the mean remaining fraction results calculated by two methods
in four seasons. n represent the number of samples used in statistics.

In order to analyze the underlying reason, first, we checked the equation used to
calculate $F_{iso}$ and $F_{raindrop}$. We noted that in both methods, RH is an important parameter
to compute the remaining ratio. In the equation for computing $F_{iso}$, the values of γ and





β are highly dependent on RH. Equally, in the $F_{raindrop}$ computing equation, RH will be
the decisive factor of evaporation intensity (E). Then, we tested the sensitivity between
$\Delta\delta^{18}O$ and RH under different $F_{iso}$ levels (Fig. S5). Our results showed, under high RH
condition (60%~90%), a little variation of $\Delta\delta^{18}O$ corresponded to a wide range of $F_{iso}$
distribution. We also noticed, under higher RH condition (above 90%), the simulated
$\Delta\delta^{18}O$ is very small, normally lower than 0.5‰. However, in reality, the $\Delta\delta^{18}O$ is
generally greater than 0.5‰. Therefore, when the actual $\Delta\delta^{18}O$ value is larger than the
theoretical value, the calculated $F_{iso}$ results will be larger than 100%, and this is in
accordance with the actual condition. Because under higher RH condition, the raindrop
evaporation ratio will decrease, and in turn the $F_r$ will appropriately increase. Moreover,
in the near-saturated air column, the raindrop is hardly evaporated.

Therefore, it is reasonable to assume that when the RH is higher, the difference
between the $F_{iso}$ and $F_{raindrop}$ will be reduced. To validate our assumption, we computed
the $F_{iso}$ and $F_{raindrop}$ by increasing RH by 20%, respectively. As expected, the mean
annual difference was highly reduced, and statistically there is no significant difference
(Fig. 6a, independent t-test, F=5.665, p=0.075). Moreover, the $F_r$ computed by those
two methods is closer to each other, while the correlation coefficient is highly increased,
and the slope is closer to 1 (Fig. 6b). For the seasonal variations of $F_r$, the larger
differences between $F_{iso}$ and $F_{raindrop}$ in spring and summer are regarding to the low RH
in these seasons, while the small difference in autumn is related to the higher RH. For
the largest difference in winter, it is most likely due to the fact that in the mass
conservation model, the diameter of raindrop used to determine the terminal velocity
of the raindrop ($v_{end}$) and the evaporation intensity (E) do not account for snowfall factor
resulting a great uncertainty in the calculation results.

**3.3.2 Sensitivity test**
In the below-cloud isotopic evaporation model (eq. 7), the two controlling factors are
the equilibrium fractionation factor (α) and the RH. As the equilibrium fractionation
factor varies with the cloud base altitude (mainly caused by the variation of
temperature), we used the different altitudes to represent the variations of α. In order
to assess the relevance of different ambient conditions for the raindrop evaporation, a
sensitivity test of $F_r$ under different altitude and RH scenarios is exhibited in Fig. S6.
With the increase of altitude, the $F_r$ is gradually decreased. It is well known that with
the increase of altitude, the raindrop falling distance will increase, and correspondingly
the falling time will be extended. As a result, more fraction of raindrops would be





evaporated in the unsaturated atmospheric columns. When the RH increases by 20%,
the atmospheric columns is near saturated, and largely decrease the evaporation
possibility of falling raindrops. Conversely, the decrease of RH will strongly increase
the evaporation proportion of falling raindrops. In addition, according to Fig. S6, the $F_r$
seems to be more sensitive to the changing of RH than that of altitude.
Comparing with the isotopic method, there are many parameters in the mass
conservation model, such as raindrop diameter, evaporation intensity, raindrop falling
velocity, resulting in the remaining fraction calculated by the mass conservation model
with larger uncertainty. Taking the $F_{iso}$ results as the benchmark, in our study, the mass
conservation method will overestimate the raindrop evaporation ratios. The
overestimation may be related to the low RH in our studying location. If we increase
the RH by 20%, there is no significant difference between the two methods. This
indicates that in high RH areas, either method could be used to calculate the $F_r$.
However, in those arid and semi-arid areas, where the RH is relatively low, and the
high latitude regions, where snowfall is frequent in winter, we need to cautiously use
the result computed by the mass conservation method. Furthermore, Graf et al (2019)
emphasized the role of the temperature structure, in particular melting layer height in
the influence on the below-cloud processes, that a higher melting layer height prolongs
the time for exchange between vapor and rain and leads to stronger equilibration and
evaporation. In the future, it is therefore promising to study the raindrop formation
heights, temperature profiles (e.g. melting layer heights), and atmospheric water vapor
isotopic profiles when considering the below-cloud processes of the raindrops.
**3.4 The characteristics of below-cloud evaporation of raindrop in Xi'an**
As the phenomenon of below-cloud evaporation is very common in arid and semi-arid
regions, to explore the information contained in the precipitation isotopic composition,
it is important to clearly know that how much of the raindrops have been evaporated
before they land on the ground. Here, we summarized the seasonal variations of $F_{iso}$
in Xi'an (Fig. 8).



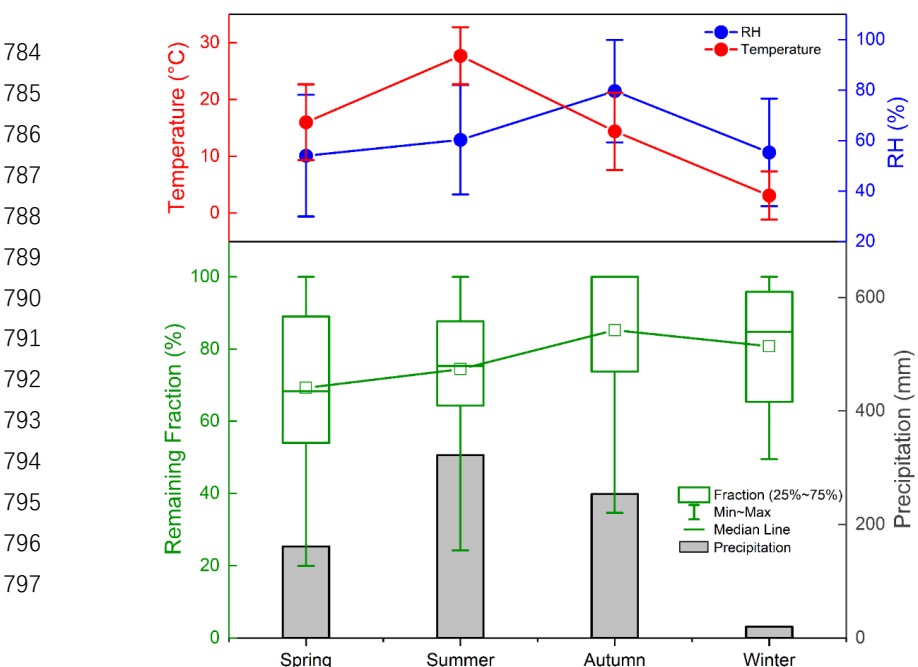

Figure 8 The variations of temperature, relative humidity, precipitation amount and mean remaining fraction of evaporated raindrops in four seasons in Xi'an

By seasonally dividing the precipitation isotopic composition on the ΔdΔδ-diagram, it showed that samples collected in spring and summer dominate the evaporation phase, reflecting a stronger evaporation influence, while most of the winter precipitation and part of autumn precipitation monopolize the cloud signal phase indicating a weak or no below-cloud evaporation on these samples (Fig. S7). In addition, part of the autumn samples, of which the below-cloud evaporation and cloud-based isotopic exchange tends to achieve a complete equilibrium state is distributed in the equilibration phase (Fig. S7).

The mean raindrop evaporation rate is highest in spring and lowest in autumn based on two-year data (Fig. 8). The seasonal variation of $F_{iso}$ basically followed the trend of seasonal variation of RH. Although the precipitation amount is highest in the summer, the temperature is extremely high and RH is relatively low, which causes the high evaporation rate in summer. In winter, the low evaporation rate may be related to the precipitation type, because snowfall is the main deposition type in this season.

**4 Conclusions**

The below-cloud processes of precipitation are complex, variable, and influenced by





many factors, especially in the arid and semi-arid regions. Previously, below-cloud evaporation is the most well-studied post-condensation process with the aid of the slope of LMWL and d-excess of precipitation. In comparison, other below-cloud processes, such as the equilibration between the raindrop and ambient vapor, have paid less attention in different rain types. In this study, based on the two-year precipitation data collected in Xi'an, we systematically analyze its below-cloud processes, and get the following main conclusions:

1. In Xi'an, the precipitation isotopic signals mainly record the information of water vapor isotopic composition, but the signals could be changed by the below-cloud evaporation effect. This reminds us to be cautious in using precipitation isotopic compositions to study the hydrological cycle and climate changes in the arid and semi-arid regions.

2. Our work validates the general applicability of the ΔdΔδ-diagram. Although there is a difference in timescale between Graf's et al. (2019) study (intra-event) and ours (per-event), by presenting our per-event precipitation isotopic results on the ΔdΔδ-diagram, the below-cloud processes and their effects on the isotopic composition of vapor and precipitation can be clearly visualized. In Xi'an, the below-cloud evaporation is the main process during the raindrops falling. Snowfall samples are less influenced by the below-cloud processes and preserve their initial water vapor information. Hence, our results strengthen the reliability of using ice core to reconstruct the paleoclimate, paleoenvironment, and paleohydrology in the cold area. The different Δd/Δδ slopes of rainfall and snowfall may be related to the precipitation types.

3. Compared with the isotopic method, the evaporation rate computed by the mass conservation model is overestimated. The relative humidity is the main controlling factor in computing the remaining fraction of raindrops below-cloud evaporation. Due to more uncertain parameters in the mass conservation model, such as raindrop diameter, evaporation intensity, raindrop falling velocity, and no consideration of precipitation type, it is more suitable to use the isotopic model to calculate the remaining fraction of evaporated raindrops.

4. In Xi'an, the evaporation rates are higher in spring and summer, and lower in autumn and winter, and this is related to the variation of local RH.



**Data availability**

The datasets can be obtained from the TableS1.

**Author contribution**

Meng Xing and Weiguo Liu designed the experiments, interpreted the results, and prepared the manuscript with contributions from all co-authors. Meng Xing and Jing Hu analyzed the precipitation and water vapor samples. Jing Hu maintained the experimental instruments.

**Competing interests**

The authors declare that they have no conflict of interest.

**Acknowledgment**

This work was supported by Science Foundation of China (No. 42177093), West Light Foundation of The Chinese Academy of Sciences, and China scholarship council. The authors would like to thank Mr. Xijing Cao for helping to collect precipitation samples.

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
