# Peer review of "A comparison of two methods to quantitatively evaluate the effect of below-cloud evaporation on the precipitation isotopic composition in the semi-arid region of the Chinese Loess Plateau"

_Atmospheric Chemistry and Physics, 2022_

## Referee Comment (RC1)

**Review on ACP Manuscript No. ACP-2022-576**

Xing et al., A comparison of two methods to quantitatively evaluate the effect of below-cloud evaporation on the precipitation isotopic composition in the semi-arid region of the Chinese Loess Plateau

Xing et al. use measurement data for two years derived in the Chinese Loess Plateau and investigate the impact of below-cloud evaporation processes on the isotopic composition. Thereby, they compare two methods. Their comparison shows that the equilibrium method can be successfully used to predict the ground-level water isotopic composition from precipitation isotopes in semi-arid climates, especially in winter. On the other hand, the below-cloud evaporation calculated from the mass conservation equation would lead to an overestimation in semi-arid and arid regions.

Due to the poor language and the extreme length of the paper it is hard to follow what actually has been done and how. Also the purpose of the study is rather confusing. Title and abstract focus on a comparison of two methods and then in the result section much more is shown and the methods are somewhere compared without clearly stating this. Therefore, this manuscript needs major revisions before it can be published. I would strongly encourage the authors to considerably shorten their manuscript and focus on the essential results and clearly describe these.

**General comments:**

1. The abstract is quite confusing and should be revised that the content, methods and results of this study become more clear.

2. The introduction is with four pages too long and should be significantly shortened by 1-2 pages.

3. It generally needs to be more clearly stated (abstract and introduction) which two methods are used and what the differences between these methods are (see specific comments below).

4. The method section is also somewhat lengthy and should be shortened. Some of the descriptions and information could be provided in an appendix.

5. Also the Result section is very lengthy and it becomes not clear where you actually compare the two methods and how you come to the conclusion that ones is overestimating the below-cloud processes while the other one is underestimating these.

6. A thorough language check should be made before re-submission of the manuscript. Much of the questions and comments arise due to a poor language.

**Specific comments:**

P2, L33-34: How does the below-cloud alter the isotopic composition? Why does this lead to a misinterpretation of the signal? This is not clear. If you want to start your abstract like this you have to be more precise and provide more explanations.

P2, L42-44: Why is this important to be mentioned in the abstract? What information does one get form this value range? What does it mean?

P2, L47: What relationship is considered here? You should explicitly state what this diagram is, i.e. that you use the relation between d-excess and the isotopes.

P2, L45 and L53: Here you mention the methods, but do not introduce them properly as method 1 and 2. Further, in the abstract it should be clearly stated, as it is done in the manuscript title, that you are comparing two methods. Furthermore, the characteristics of each method should be shortly described.

P2, L54: What is the "remaining fraction of raindrop mass"? What does the reader learn from this parameter and the numbers given?

P2, L59: Which methods have been designed? Does that mean you have the methods developed yourself? Aren't these established methods that are used and just compared?

P3, L69-70: Since this sentence at the beginning of the abstract is rather misleading I would suggest to move this sentence to L82 and start with "Thus, …….". The original sentence starting in line 82 could then start directly with "However" (and skip "however" it in the middle of the sentence), thus that it reads "However, due to the ……." then we first paragraph makes more sense and is more logical structured.

P3, L93: The relation to climate change is not clear, especially in the frame of your study. You are using two years of data. With this set of data you can hardly derive any results on climate change. Thus, climate change should be deleted in this sentence.

P95-96: I still do not get the point. All processes that isotopes are affected by are manifested somehow in the isotopic composition. To understand the isotopic composition the processes have to be disentangled and for this certain methods can be used. Isn't then the main purpose of this study to just quantify how large the contribution from below-cloud evaporation is on the isotopic composition?

P107-109: The relationship between isotopic composition ($2H1H16O$ and $1H218O$) and isotopic ratio (del2H and del18O) has not been made clear and you should carefully check your text when you refer to the isotopic composition and when to the isotopic ratio.

P4, L111: Also here your statement is not entirely clear. You state that the non-equilibrium effect cause a decrease of d-excess, but how is it with the equilibrium effects? Do these cause and increase in d-excess? You actually write it two sentences later. For better readability this sentence should be moved higher up.

P6, L174ff: This section is already too detailed on the methods and should thus be moved to the method section.

P6, L176 and L183: Which model? Before you stated you are using two methods, thus I think you mean here rather method.

P6, L190 and 192: Here again you speak about a model, but later and before these were methods and not models.

P6, L197: You still have not explained what the $\Delta d \Delta \delta$-diagram is and what it is used for.

P8, L269: What do you mean with high-precision model? This is an instrument. You rather mean a high precision version of the instrument? Or do you mean measured with a high precision?

P8, L277: What do you mean with "to the scale of two standard material VSMOW-GISP"? What is the abbreviation VSMWOW-GISP standing for? Do you mean with "two"? Two standard deviations, thus two sigma?

P10, L328: What is the fourth quadrant of the $\Delta$d$\Delta\delta$-diagram? What kind of separation can be made from this diagram can be made? This hasn't been explained.

P11, L358: Why do you calibrate to VSMOW-GISP? Why do you need to do this?

P14, L472: You still have nowhere clearly stated which two methods you are using. Using the term model always before causes even more confusion.

P15, L511: How do your derive this number? Is this derived from your study or known from other sources? In the former case more explanation is needed, in the latter case a reference should be added.

P15, L513: How is this value for the cloud base justified? More information needs to be provided.

P19, Figure 5: Legend for which are the snow samples and which are the rain samples should be added (e.g. at the lower left corner of the plot or you make one for all suplots on the right bottom of the figure).

P20, L642: What are "intra-event" and "per-event" samples? This needs to be more explanations to understand that the differences between the Graf et al. and your data set are.

P21, L661: Are you here comparing the two methods? If yes, what has exactly be done before. Does the title and the introduction then correctly describe what you are actually showing in this study?

Generally: Due to the length of the manuscript and the large amount of figures (including supplement) I lost track of what the purpose of this study is. It seems not to be solely the comparison of the two methods used in this study.

P26, l828: Remove "climate change" since the connection to climate change does not become clear from your study.

P848: This is not a good last sentence for the paper. You should move this bullet point higher up, thus first summarize the results for X'ian and the general results.

**Technical comments:**

P2, L42: Add "isotope" after precipitation and add "water vapour isotopes" before d2H to be more clear.

P2, L61: signal → composition

P2, L53: Check sentence. Is "while" correct here? If this latter part of the sentence is an explanation then it should rather read "since". Otherwise, the sentence in itself is not correct and needs to be rephrased.

P3, L69: Is "greatly" correct here? It should rather read "most"

P3, L82: Change sentence as follows "….itself is an important part of the hydrological cycle."

P3, L89: Add "The" → "The Chinese Loess Plateau…….."

P3, L93: climate changes → changes in climates

P3, L93: distorted → affected

P4, L112: FISHER → Fisher

P4, L114 and L123: Make a line break here and start a new paragraph.

P4, L127: lose → lost

P6, L164: Delete "As a creative work".

P6, L170: add "that" before experience and delete "effect"

P5, L171: need to do → need to be done

P6, L194: Here we have measured → Here we use measurements

P6, L206: Meanwhile should rather be "Thus" or "Therefore"

P7, L219: reported by many studies in → reported in many studies for

P7, L242: add "site" after measurement

P8, L268 and L280: by Picarro → with a Picarro

P8, L281: instead of "model" you should write "version of the instrument".

P9, L299-300: Sentence not correct. Please check and rephrase.

P9, L307: China → Chinese

P10, L334-337: Sentence not correct. Please check and rephrase.

P10, L346-347: Sentence not correct. What do you mean with "were used to calculate the average to be recognized as the δ-value at the measured humidity"? Please rephrase.

P11, L366: representative → representatives

P11, L367: two-year study → two years of measurements

P11, L368: Add "event" after rainfall

P11, L373: add measurements, so that it reads "isotopic composition measurements"

P11, L373-374: Second part of the sentence not clear. Please rephrase.

P11, L377: Move "in summer and autumn" at the begin of the sentence.

P12, L396: The second d-excess should have the indice "gr-v"

P12, L417: By isotope? Do you mean by the isotope method?

P12, L418: Add "that" so that it reads "suggested that".

P12, L419: Add "that" so that it reads "mass that remained".

P13, L430: Model? It should rather be "method".

P13, L437: have → has

P13, L443: reaching ground raindrop → raindrops reaching the ground-l

P13, L450: model → method

P14, L467: "respectively" obsolete and change "used" to "use"

P15, L400: Add "value" after "range"

P15, L409: Start a new sentence after "Figure 3a": "As expected……….."

P17, L548: Sentence incomplete. Unsaturated what? Conditions? Environment?

P18, L584-585: from through → by

P18, L594: on Graf et al. ….→ on the by Graf et al. …...

P18, L600: by combined with → in combination with

P19, L607: Add "one" so that it reads "use one single physical variable".

P19, L619: Add "to be" so that it reads "to be distributed".

P20, L646: Here you write $\Delta d \Delta \delta$ with a slash in between, but before it was written without a slash. This should be done one or the other way consequently throughout the manuscript.

P20, L658: to do → to be done

P21, L664: Limited → limited

P21, L668: used → use

P21, L676: to do → to be done

P21, L679: reaming → remaining

P21, Figure 6, right panel, x-label: remaing → remaining

P22, L694: in statistics → in the statistics

P22, L705: pointed that → pointed out that

P22, L710: Add what is shown in red and what in blue in the caption.

P23, L715: Delete "computing"

P23, L717: Add "that" so that it read "Our results showed that".

Reference list: Should be checked thoroughly so that the citation style is the same for all references and that chemical species names are printed correctly.

---

## Referee Comment (RC2)

Review acp-2022-576

**A comparison of two methods to quantitatively evaluate the effect of below-cloud evaporation on the precipitation isotopic composition in the semi-arid region of the Chinese Loess Plateau**

**Meng Xing, Weiguo Liu, Jing Hu, and Zheng Wang**

The manuscript "A comparison of two methods to quantitatively evaluate the effect of below-cloud evaporation on the precipitation isotopic composition in the semi-arid region of the Chinese Loess Plateau" presents a two-year dataset of the stable water isotope composition of water vapour and precipitation collected in Xi'an in northwestern China. The authors apply two methods to calculate the remaining precipitation fraction after below-cloud evaporation. The results from these two methods are compared and differences are discussed using sensitivity studies. This study presents a valuable dataset of paired precipitation and water vapour measurements that are well described and characterised. But it is difficult to follow the analysis of the below-cloud effects. Further, the purpose of this study is not fully clear after reading the manuscript.

**General comments:**

- The two methods used to calculate the below-cloud effect have to be better introduced: What assumptions are needed? How did you choose the unknown values? What are the differences between the methods? Why are you using the isotopic method as a benchmark and not the mass conservation method?
- Due to the large number of assumptions needed in the below-cloud evaporation models, the sensitivity analysis has to be more prominent. Further, the different model simulations have to be introduced better. Currently, it is difficult to understand the difference between the models and the different simulations.
- The manuscript is often repetitive. References to the methods are repeated in many places and the introduction to the sections and paragraphs are too general without leading towards the main topics of the paragraphs.

**Specific comments:**

- Introduction: this section is very long, consider shortening it.
- line 269: model -> do you mean mode? (same on line 281)
- line 269-271: For how long did you measure each liquid injection? Did you apply a drift correction?
- Line 287: "measured by a CTC Analytics autosampler...", do mean that the samples were injected using the autosampler? I expect the measurements to be done by the laser spectrometer.
- Lines 311-331: This paragraph seems out of place as it discusses a method that hasn't been introduced yet.
- Lines 334-335: The first part of this sentence is difficult to understand, consider reformulating.
- Lines 344-347: For how long did you measure each liquid injection? Did you apply a drift correction?
- Lines 349-350: The humidity-isotope dependency as shown in S1a and S1b shows a dependency on the isotopic composition of the standards as reported by Weng et al. (2020). For example in S1b, LS-1 shows a decrease in $\Delta\delta2H$ with decreasing humidity while LS-3 shows an increase with decreasing humidity. The linear calibration function of $\delta2H$ does not take this into account. Therefore, the humidity-isotope calibration functions (eq 1 and 2) should be reconsidered to include this isotope-dependency.
- Line 355: "$\delta_{\text{humidity calibration}}$ is the calibrated data for water vapor stable isotope" do you mean "... is the *humidity-dependency corrected* data.."?
- Line 266: representative -> do you mean "representativeness"?
- Line 373-374: "of which 100 precipitation samples have corresponding daily average water vapor isotopic results" Does this mean you compared the precipitation isotopes with daily average water vapour data? If yes, this would mean that you don't always compare the same time periods with each other. To compare precipitiation and water vapour, the water vapour isotopic composition should be averaged over the time period of the precipitiation event.
- 388: "various" -> do you mean "different"?
- Lines 389-397: $\delta_{\text{pv-eq}}$ and d-excess$_{\text{pv-eq}}$ are defined twice in different ways. On lines 390-391, it says that these variables represent the equilibrium vapour from the precipitation samples, on lines 396-397, it says that they represent the water vapour composition at cloud base.
- Line 396: d-excess$_{\text{pv-eq}}$ and d-excess$_{\textbf{pv-eq}}$ -> d-excess$_{\text{pv-eq}}$ and d-excess$_{\textbf{gr-v}}$

- Sections 2.5.2 and 2.5.3: These two section introduce the two methods used to calculate the below-cloud effect. After reading these two section, I still didn't fully unterstand which assuptions were made. I think that part of the Appendix A has to be mentioned in 3.5.2 (e.g. how the isotopic composition of preciptiation at the cloud base is estimated). A conceptual schematic of the properties (and assumptions) between cloud base and ground and how they differ between the two methods might help to understand the two models better.
- Section 2.5.3 The first and last paragraph repeat a lot of information already mentioned earlier in the manuscript.
- Line 508-509: "the corresponding day's water vapour isotopic composition" why do you not compare the per-event mean water vapour isotopic composition? The water vapour isotopic composition changes strongly pre-, intra- and post-event (e.g. Aemisegger et al. 2015). If you average over the full day instead of the precipitation period, the average water vapour isotopic composition can differ strongly.
- Line 513: Is there any seasonal cycle in the mean cloud base at your measurement location?
- Line 525-526: "equilibrium prediction" -> do you mean "equilibrium water vapour from preciptiation"?
- Line 534: How is $\delta^{18}O_{v\text{-}eq(1500m)}$ calculated?
- Lines 535-542: The sentence "our results indicate that it is possible to derive the isotope composition of atmospheric water vapor based on that of the precipitation in the semi-arid area." seems to contradict "It is worth noting that we do not propose to extract the water vapor isotope time series from precipitation data." What is your message here?
- Line 555: d-excess is 0‰ during equilibrium fractionation only at temperatures around 20°C.
- Lines 560-561: Kinetic/non-equilibrium fractionation should be introduced earlier, and used consistently.
- Line 562-563: "Therefore, during the moisture transportation, the water vapor d-excess may be modified." What do you mean with *moisture transportation*? Diffusion or large-scale advection? It is important to specify the scale of the process.
- Line 584-585: "...from through..." -> this is a repetition.
- Lines 591-593: Can you be more specific about what is the *new* learning from your results? Is has been known before, that snow does not interact strongly with sourrounding water vapour below the cloud base during its fall (e.g. Gedzelman and Arnold, 1994).
- Line 603: $\Delta^2H_v$ -> do you mean $\Delta\delta^2H_v$? This notation isn't used consistently in the manuscript.
- Line 603-621 and Fig.5: The discussed connection between meteorological conditions and the isotopic values is difficult to see in these figures. Boxplots instead of scatterplots might work better.
- Line 615-617: I don't understand this sentence. Are you referring to the temperature dependency of equilibrium fractionation?
- Line 670: "below-unsaturation" -> what does that mean?

**Figures:**

- Fig. 1: The labels of the subfigures are very small
- Fig.4: The linear fit does not fit well to the data. Why do try to find a linear fit for snow and rain together? As these hydrometeors are influenced by different processes while falling, it is unlikely, that they lie on the same line in this diagram.
- Fig.5: label text is very small. The dark red of very high values (e.g. temperature) is difficult to see.

**References:**

Aemisegger, F., Spiegel, J. K., Pfahl, S., Sodemann, H., Eugster, W., and Wernli, H., 2015: Isotope meteorology of cold front passages: A case study combining observations and modeling, Geophys. Res. Lett., 42, 5652–5660, doi:10.1002/2015GL063988.

Gedzelman, S. D. and R. Arnold, 1994: Modeling the isotopic composition of precipitation. J. Geophys. Res. Atmos., 99, 10455–10471, doi:10.1029/93JD03518.

Weng, Y., Touzeau, A., and Sodemann, H., 2020: Correcting the impact of the isotope composition on the mixing ratio dependency of water vapour isotope measurements with cavity ring-down spectrometers, Atmos. Meas. Tech., 13, 3167–3190, doi: 10.5194/amt-13-3167-2020.

---

## Author Comment (AC1)

Reviewer #1

1. The abstract is quite confusing and should be revised that the content, methods and results of this study become more clear.

Thanks for your suggestion. We have revised the abstract, and now it reads "When the hydrometeor falls from the in-cloud saturated environment towards the ground, especially in the arid and semi-arid regions, the below-cloud processes could heavily alter the precipitation isotopic composition through equilibrium and non-equilibrium fractionations, and accounts for the misinterpretation of precipitation isotopic signal if these processes cannot be properly identified. To correctly understand the environmental information contained in the precipitation isotopes, qualitatively analyzing the below-cloud processes and quantitatively calculating the below-cloud evaporation effect are becoming very important. Here, based on a two-year synchronous observations of precipitation and water vapor isotopes in Xi'an, we compiled a set of effective methods to systematically evaluate the below-cloud evaporation effect on local precipitation isotopic composition. The $\Delta d\Delta\delta$-diagram shows the isotopic differences ($\delta^2 H$, d-excess) of the precipitation equilibrated vapor relative to the observed vapor, in which the equilibration and evaporation could lead to different pathways in the two-dimensional phase space. By using $\Delta d\Delta\delta$-diagram, our data show that evaporation is the major below-cloud process, while snowfall samples retain the initial cloud signal because of less isotopic exchange between vapor and solid phases. To quantitatively characterize the influence of below-cloud evaporation on precipitation isotopic composition, here, we chose two methods: one is based on the raindrop's mass change during its falling (hereafter referred to as method 1); another is to directly calculate the precipitation isotopic variations from the cloud base to the ground (hereafter referred to as method 2). By comparison, we found that there are no statistical differences between the two methods in evaluating the evaporation effect on $\delta^2 H_p$, except for snowfall events. The slope of evaporation proportion and difference in $\delta^2 H$ ($F_i/\Delta\delta^2 H$) is a little larger in method 1 (1.0 ‰/%) than in method 2 (0.9 ‰/%). Additionally, both methods indicate that the raindrops are weakly evaporated in autumn, and heavily evaporated in spring. Through the sensitivity test, relative humidity is the most sensitive parameter in both, while the variations of temperature show different effects on the two methods. Therefore, following our methods, the diagnosis of below-cloud processes and the understanding of their effects on the precipitation isotopic composition will be improved."

2. The introduction is with four pages too long and should be significantly shortened by 1-2 pages.

Thanks for your suggestion. We have shortened the introduction part to 3 pages.

3. It generally needs to be more clearly stated (abstract and introduction) which two methods are used and what the differences between these methods are (see specific comments below).

Thanks for your suggestion. We have followed your suggestion to revise the sentence. The sentence now reads "To quantitatively characterize the influence of below-cloud evaporation on precipitation isotopic composition, here, we chose two methods: one is based on the raindrop's mass change during its

falling (hereafter referred to as method 1); another is to directly calculate the precipitation isotopic variations from the cloud base to the ground (hereafter referred to as method 2). By comparison, we found that there are no statistical differences between the two methods in evaluating the evaporation effect on $\delta^2 H_p$, except for snowfall events. The slope of evaporation proportion and difference in $\delta^2 H$ ($F_i/\Delta\delta^2 H$) is a little larger in method 1 (1.0 ‰/%) than in method 2 (0.9 ‰/%).".

4. The method section is also somewhat lengthy and should be shortened. Some of the descriptions and information could be provided in an appendix.
Following your suggestion, we have shortened the method section, and moved some descriptions and information to the supplemental material as appendixes. Please see the revision.

5. Also the Result section is very lengthy and it becomes not clear where you actually compare the two methods and how you come to the conclusion that ones is overestimating the below-cloud processes while the other one is underestimating these.
Thanks for your suggestion. We have rewritten the result section, and deleted the redundant content. Now, according to your suggestion, we separated the methods into 1 and 2, and explicitly compared them in Section 3.3. In the revision, we just compared the two methods, pointed out flaws, and did not evaluate which one overestimates the results and which one underestimates the results.

6. A thorough language check should be made before re-submission of the manuscript. Much of the questions and comments arise due to a poor language.
We have seriously revised our manuscript following your suggestions. In addition, we have checked the English grammar and readability of the manuscript. Now, we believe it has reached the quality for publishing.

P2, L33-34: How does the below-cloud alter the isotopic composition? Why does this lead to a misinterpretation of the signal? This is not clear. If you want to start your abstract like this you have to be more precise and provide more explanations.
Thanks for your suggestion. We have rephrased the sentence to "When the hydrometeor falls from the in-cloud saturated environment towards the ground, especially in the arid and semi-arid regions, the below-cloud processes could heavily alter the precipitation isotopic composition through equilibrium and non-equilibrium fractionations, and accounts for the misinterpretation of precipitation isotopic signal if these processes cannot be properly identified."

P2, L42-44: Why is this important to be mentioned in the abstract? What information does one get from this value range? What does it mean?
Yes, you are right. After considering your question, we have deleted this sentence.

P2, L47: What relationship is considered here? You should explicitly state what this diagram is, i.e. that you use the relation between d-excess and the isotopes.

Thanks for your suggestion. Now the sentence reads "The $\Delta d\Delta\delta$-diagram shows the isotopic differences ($\delta^2H$, d-excess) of the precipitation equilibrated vapor relative to the observed vapor, in which the equilibration and evaporation could lead to different pathways in the two-dimensional phase space. By using $\Delta d\Delta\delta$-diagram, our data show that evaporation is the major below-cloud process, while snowfall samples retain the initial cloud signal because of less isotopic exchange between vapor and solid phases."

P2, L45 and L53: Here you mention the methods, but do not introduce them properly as method 1 and 2. Further, in the abstract it should be clearly stated, as it is done in the manuscript title, that you are comparing two methods. Furthermore, the characteristics of each method should be shortly described.
Thanks for your suggestion. We have followed your suggestion to revise the sentence. The sentence now reads "To quantitatively characterize the influence of below-cloud evaporation on precipitation isotopic composition, here, we chose two methods: one is based on the raindrop's mass change during its falling (hereafter referred to as method 1); another is to directly calculate the precipitation isotopic variations from the cloud base to the ground (hereafter referred to as method 2). By comparison, we found that there are no statistical differences between the two methods in evaluating the evaporation effect on $\delta^2H_p$, except for snowfall events. The slope of evaporation proportion and difference in $\delta^2H$ ($F_i/\Delta\delta^2H$) is a little larger in method 1 (1.0 ‰/%) than in method 2 (0.9 ‰/%)."

P2, L54: What is the "remaining fraction of raindrop mass"? What does the reader learn from this parameter and the numbers given?
To make our expression more clear, we have revised the sentence to "Through the sensitivity test, relative humidity is the most sensitive parameter in both, while the variations of temperature show different effects on the two methods", and deleted the numbers.

P2, L59: Which methods have been designed? Does that mean you have the methods developed yourself? Aren't these established methods that are used and just compared?
Yes, you are right. Here, we just compiled the established methods to qualitatively analyze the below-cloud processes that the raindrops are experienced and quantitatively calculate the below-cloud evaporation ratio. Thus, we have rephrased the sentence to "Here, based on a two-year synchronous observations of precipitation and water vapor isotopes in Xi'an, we compiled a set of effective methods to systematically evaluate the below-cloud evaporation effect on local precipitation isotopic composition."

P3, L69-70: Since this sentence at the beginning of the abstract is rather misleading I would suggest to move this sentence to L82 and start with "Thus, ……". The original sentence starting in line 82 could then start directly with "However" (and skip "however" it in the middle of the sentence), thus that it reads "However, due to the …….." then we first paragraph makes more sense and is more logical structured
Following your suggestion, we have revised this paragraph, and it reads "For the paleoenvironment, the isotopic signals of precipitation recorded in ice cores

(Thompson et al., 2000; Yao et al., 1996), tree rings (Liu et al., 2004; Liu et al., 2017b), speleothems (Cai et al., 2010; Tan et al., 2014), and leaf wax of loess-paleosol deposits (Wang et al., 2018b) and lake sediments (Liu et al., 2017a, 2019) could be used to reconstruct the information of temperature, precipitation, and hydrological regimes in geologic history, as it had participated into the formation or growth of these geological archives. For the modern environment, the isotopic ratios of precipitation could be used to quantitatively constraint the water vapor contribution from the end-members of advection (Peng et al., 2011), evaporation (Sun et al., 2020; Wang et al., 2016a), transpiration (Li et al., 2016a; Zhao et al., 2019), and even anthropogenic activities (Fiorella et al., 2018; Gorski et al., 2015; Xing et al., 2020), as itself is an important part of the hydrological cycle. Thus, the hydrogen and oxygen isotopes of precipitation are one of the most important tools to trace the hydrological cycle and climate change (Bowen et al., 2019; Gat, 1996). However, due to the limitations in sampling and isotopic fractionation theories, there remains large uncertainty (i.e., the remaining fraction of below-cloud evaporation, the moisture recycling ratio, water molecules exchange between the droplet and ambient air, etc.) in deciphering the information contained in precipitation by using hydrogen and oxygen isotope ratios (Bowen et al., 2019; Yao et al., 2013)".

P3, L93: The relation to climate change is not clear, especially in the frame of your study. You are using two years of data. With this set of data you can hardly derive any results on climate change. Thus, climate change should be deleted in this sentence.
Yes, you are right. We have deleted the sentence in the revision.

P95-96: I still do not get the point. All processes that isotopes are affected by are manifested somehow in the isotopic composition. To understand the isotopic composition the processes have to be disentangled and for this certain methods can be used. Isn't then the main purpose of this study to just quantify how large the contribution from below-cloud evaporation is on the isotopic composition?
Yes, you are right. The part of content looks a little redundant, we have deleted it in the revised manuscript.

P107-109: The relationship between isotopic composition ($2H1H16O$ and $1H218O$) and isotopic ratio ($del2H$ and $del18O$) has not been made clear and you should carefully check your text when you refer to the isotopic composition and when to the isotopic ratio.
Thanks for your suggestion. We have checked our text, and revised our description. When the isotope without "δ", we used the isotope ratio to express; and when the isotope with "δ", we used isotopic composition to express.

P4, L111: Also here your statement is not entirely clear. You state that the non-equilibrium effect cause a decrease of d-excess, but how is it with the equilibrium effects? Do these cause and increase in d-excess? You actually write it two sentences later. For better readability this sentence should be moved higher up.
Thanks for your suggestion. We have revised the sentence, and now it reads "The equilibrium fractionation would not change the d-excess, while the nonequilibrium diffusional process would result in a decrease of d-excess in rain (Fisher, 1991; Merlivat and Jouzel, 1979). Additionally, the slope of the local meteoric water line (LMWL) has also been widely used as a metric to infer the below-cloud evaporation effect according to the theory of water isotope equilibrium fractionation (Chakraborty et al., 2016; Putman et al., 2019b; Wang et al., 2018a). Generally, the LMWL's slope is approximately equal to 8.0 belonging to equilibrium fractionation and that is lower than 8.0 pointing to a non-equilibrium fractionation, such as the re-evaporation of raindrops."

P6, L174ff: This section is already too detailed on the methods and should thus be moved to the method section.
Thanks for your suggestion. We have deleted this section.

P6, L176 and L183: Which model? Before you stated you are using two methods, thus I think you mean here rather method.
We have deleted this sentence. And in the revision, we have changed "model" to "method".

P6, L190 and 192: Here again you speak about a model, but later and before these were methods and not models.
We have deleted this sentence. And in the revision, we have changed "model" to "method".

P6, L197: You still have not explained what the $\Delta d\Delta \delta$-diagram is and what it is used for.
We have explained the $\Delta d\Delta \delta$-diagram in the introduction, and it reads "Recently, Graf et al. (2019) provided a new interpretive framework to directly separate the convoluted influences on the stable isotopic composition of vapor and precipitation according to the theoretical fractionation processes, especially the influences of equilibration and below-cloud evaporation. The axes of the new diagram consist of the differences, $\Delta \delta^2 H$ and $\Delta d$, between the isotopic composition of precipitation equilibrate vapor and near-surface vapor, namely $\Delta \delta \Delta d$-diagram."

P8, L269: What do you mean with high-precision model? This is an instrument. You rather mean a high precision version of the instrument? Or do you mean measured with a high precision?
Yes, you are right. We have revised this sentence to "The precipitation samples were measured with a Picarro L2130-i (serial number HIDS 2104) wavelength-scanned cavity ring-down spectrometer at a high-precision mode."

P8, L277: What do you mean with "to the scale of two standard material VSMOW-GISP"? What is the abbreviation VSMWOW-GISP standing for? Do you mean with "two"? Two standard deviations, thus two sigma?
VSMOW and GISP are two international standards, and they are the abbreviations of Vienna Standard Mean Ocean Water and Greenland Ice Sheet Precipitation. To prevent misunderstanding, we have revised this sentence to "calibrated to the scale of two international standards VSMOW(Vienna Standard Mean Ocean Water)-GISP(Greenland Ice Sheet Precipitation)."

P10, L328: What is the fourth quadrant of the $\Delta d\Delta\delta$-diagram? What kind of separation can be made from this diagram can be made? This hasn't been explained.

Thanks for your suggestion, we have added the category of quadrant in Figure 4. Please refer to Figure 4 in the revision.

P11, L358: Why do you calibrate to VSMOW-GISP? Why do you need to do this?

Normally, the measured isotope compositions need to calibrate to a set of international standards. In this way, the researchers can compare their data with each other.

P14, L472: You still have nowhere clearly stated which two methods you are using. Using the term model always before causes even more confusion.

Many thanks. According to your suggestion, we have used method 1 and method 2 to represent our two methods, in addition, we have deleted the term "model" in the revised manuscript. Now, we believe our statement is more clear.

P15, L511: How do your derive this number? Is this derived from your study or known from other sources? In the former case more explanation is needed, in the latter case a reference should be added.

We derived this number based on the correlation coefficient in Fig. 3a, that is, the $R^2$ is equal to 0.70. Following your suggestion, we have revised the sentence to "As expected, they show a significant positive correlation ($R^2$=0.70, p<0.01), and thus the water vapor isotopic composition can explain 70% of the variation of precipitation isotopic composition."

P15, L513: How is this value for the cloud base justified? More information needs to be provided.

Here, we compared the precipitation-equilibrated water vapor isotopic composition at the ground level with the observed one. Therefore, we revised the sentence to "Further, we used the measured precipitation isotopic composition to deduce the water vapor isotopic composition at the ground level according to the liquid-vapor equilibrium isotope fractionation, and compared it with observed water vapor in Fig. 3b."

P19, Figure 5: Legend for which are the snow samples and which are the rain samples should be added (e.g. at the lower left corner of the plot or you make one for all suplots on the right bottom of the figure).

In the revised manuscript, we have deleted the Figure 5.

P20, L642: What are "intra-event" and "per-event" samples? This needs to be more explanations to understand that the differences between the Graf et al. and your data set are.

Thanks for your suggestion, we have revised our statement to "It should be noted that the slope of Graf's et al. (2019) is based on intra-event samples (from the start to the end of precipitation, each interval of 10 min to collect one sample), while ours is on per-event samples (only collect one sample in each precipitation event)."

P21, L661: Are you here comparing the two methods? If yes, what has exactly be done before. Does the title and the introduction then correctly describe what you are actually showing in this study? Generally: Due to the length of the manuscript and the large amount of figures (including supplement) I lost track of what the purpose of this study is. It seems not to be solely the comparison of the two methods used in this study.

Thanks for your suggestion. After reading the full text with many times, we have changed the title to "A set of methods to evaluate the below-cloud evaporation effect on local precipitation isotopic composition: a case study in Xi'an, China."

In fact, our manuscript consists of two parts: one is to use the $\Delta d \Delta \delta$-diagram to qualitatively identify the below-cloud processes in our site, and another is to compare two methods that are used to quantitatively evaluate the below-cloud effect on the local precipitation isotopic composition. In the revised manuscript, we have reorganized the structure. Now, I think the main topics of this study have become more clear.

P26, l828: Remove "climate change" since the connection to climate change does not become clear from your study.

Thanks for your suggestion, we have removed this statement in your revision.

P848: This is not a good last sentence for the paper. You should move this bullet point higher up, thus first summarize the results for Xi'an and the general results.

Thanks for your suggestion, we have moved this bullet to the first one.

P2, L42: Add "isotope" after precipitation and add "water vapour isotopes" before d²H to be more clear.

Thanks, we have added "isotope" and "water vapor isotopes" at the proper positions.

P2, L61: signal → composition

Have done.

P2, L53: Check sentence. Is "while" correct here? If this latter part of the sentence is an explanation then it should rather read "since". Otherwise, the sentence in itself is not correct and needs to be rephrased.

Thanks for your suggestion. We have deleted this sentence in the revision.

P3, L69: Is "greatly" correct here? It should rather read "most"

Thanks for your suggestion. We have revised this word to "most".

P3, L82: Change sentence as follows "….itself is an important part of the hydrological cycle."

Following your suggestion, we have revised this sentence to "as itself is an important part of the hydrological cycle".

P3, L89: Add "The" → "The Chinese Loess Plateau…….."
Have done.

P3, L93: climate changes → changes in climates
Have done.

P3, L93: distorted → affected
Have done.

P4, L112: FISHER → Fisher
Have done.

P4, L114 and L123: Make a line break here and start a new paragraph.
Have done.

P4, L127: lose → lost
Have done.

P6, L164: Delete "As a creative work".
Have done.

P6, L170: add "that" before experience and delete "effect"
Have done. Now it reads "Although the ΔδΔd-diagram gives us a new guideline to more accurately identify the below-cloud evaporation, Graf's et al. (2019) work was only tested on a cold frontal rain event during a short time, and hence more works need to be done for validating the general applicability of their framework.".

P5, L171: need to do → need to be done
Have done.

P6, L194: Here we have measured → Here we use measurements
Have done.

P6, L206: Meanwhile should rather be "Thus" or "Therefore"
Have done.

P7, L219: reported by many studies in → reported in many studies for
Have revised. Now it reads "The notable below-cloud evaporation effect has been reported in many studies for this area".

P7, L242: add "site" after measurement
Have done.

P8, L268 and L280: by Picarro → with a Picarro

Have done.

P8, L281: instead of "model" you should write "version of the instrument".
Thanks for your suggestion. We have revised this sentence to "The precipitation samples were measured with a Picarro L2130-i wavelength-scanned cavity ring-down spectrometer at a high-precision mode".

P9, L299-300: Sentence not correct. Please check and rephrase
Thanks for your suggestion. Now, it reads "Thus, the missing data indicate that the instrument is used for measuring liquid samples or being maintained".

P9, L307: China → Chinese
Have done.

P10, L334-337: Sentence not correct. Please check and rephrase.
Thanks for your suggestion. We have revised this sentence to "Since the water vapor concentration effect and isotopic composition dependency of the cavity ringdown spectrometer have been pointed out by many studies (e.g., Bastrikov et al., 2014; Benetti et al., 2014; Steen-Larsen et al., 2013; Weng et al., 2020), it is important to determine the isotopic composition-humidity correction response function."

P10, L346-347: Sentence not correct. What do you mean with "were used to calculate the average to be recognized as the δ-value at the measured humidity"? Please rephrase.
Thanks for your suggestion. We have rephrased this sentence to "To eliminate the memory effect, the first five injections were discarded, while the last three of eight injections were used to calculate the average δ-value at the measured humidity".

P11, L366: representative → representatives
Have done.

P11, L367: two-year study → two years of measurements
Have done.

P11, L368: Add "event" after rainfall
Have done.

P11, L373: add measurements, so that it reads "isotopic composition measurements"
Have done.

P11, L373-374: Second part of the sentence not clear. Please rephrase.
Thanks for your suggestion. We have separated it into two sentences, and it reads "In 2 years, a total of 514 days of water vapor isotopic composition measurements were carried out. For 141 precipitation samples, of which 100

precipitation samples have corresponding event-based water vapor isotopic results."

P11, L377: Move "in summer and autumn" at the begin of the sentence.
Have done.

P12, L396: The second d-excess should have the indice "gr-v"
Have done.

P12, L417: By isotope? Do you mean by the isotope method?
Yes, we have revised this sub-title to "Below-cloud evaporation calculated by isotope method".

P12, L418: Add "that" so that it reads "suggested that".
Have done.

P12, L419: Add "that" so that it reads "mass that remained".
Have done.

P13, L430: Model? It should rather be "method".
Thanks for your suggestion. We have revised the sub-title to "Below-cloud evaporation calculation: Method 1 ".

P13, L437: have → has
Have done.

P13, L443: reaching ground raindrop → raindrops reaching the ground-l
Have done.

P13, L450: model → method
Have done.

P14, L467: "respectively" obsolete and change "used" to "use"
Have done.

P15, L400: Add "value" after "range"
Have done.

P15, L409: Start a new sentence after "Figure 3a": "As expected……….."
Have done.

P17, L548: Sentence incomplete. Unsaturated what? Conditions? Environment?
Thanks for your suggestion. We have changed it to "However, due to the CLP belonging to the semi-arid area, the raindrops are likely to experience evaporation in the unsaturated environment".

P18, L584-585: from through → by
Have done.

P18, L594: on Graf et al. ….→ on the by Graf et al. …...
Have done.

P18, L600: by combined with → in combination with
Have done.

P19, L607: Add "one" so that it reads "use one single physical variable".
Thanks for your suggestion. We have done.

P19, L619: Add "to be" so that it reads "to be distributed".
Have done.

P20, L646: Here you write $\Delta d \Delta \delta$ with a slash in between, but before it was written without a slash. This should be done one or the other way consequently throughout the manuscript.
Thanks for your suggestion. We have done.

P20, L658: to do → to be done
Have done.

P21, L664: Limited → limited
Have done.

P21, L668: used → use
Have done.

P21, L676: to do → to be done
Have done.

P21, L679: reaming → remaining
Have done.

P21, Figure 6, right panel, x-label: remaing → remaining
Have done.

P22, L694: in statistics → in the statistics
Have done.

P22, L705: pointed that → pointed out that
Have done.

P22, L710: Add what is shown in red and what in blue in the caption.
Have done.

P23, L715: Delete "computing"
Have done.

P23, L717: Add "that" so that it read "Our results showed that".
Have done.

Reference list: Should be checked thoroughly so that the citation style is the same for all references and that chemical species names are printed correctly. Thanks for your suggestion. We have done this work.

Reviewer #2

**General comments:**
1. The two methods used to calculate the below-cloud effect have to be better introduced: What assumptions are needed? How did you choose the unknown values? What are the differences between the methods? Why are you using the isotopic method as a benchmark and not the mass conservation method? Thanks for your suggestion. In the revision, we used method 1 and method 2 to separately represent the two methods.
The assumption in method 1 is that the initial isotopic composition of the raindrop at the cloud base is in equilibrium with the surrounding water vapor. The assumption in method 2 is that the raindrop isotopic composition ($\delta_{cb-p}$) at the cloud base is in equilibrium with the surrounding water vapor, and the observed ground-level precipitation isotopic composition ($\delta_{gr-p}$) includes the processes of evaporation, growth, and isotopically equilibrium with the surrounding vapor. In addition, during the hydrometeors falling we assumed that there is no horizontal advection into or out of the column, and no updraft or downdraft.
For the unknown values, they are calculated according to the empirical equation. In addition, we have explicitly listed the calculation processes in supplemental materials. In the former edition, we assumed the cloud base is 1500m. Now, in the revision, we also used the empirical equation to define it:

$$Z=18400(1+\frac{T_{mean}}{273})\lg\frac{S_0}{S_{LCL}}$$

The fundamental differences between the two methods are: method 1 makes use of the mass change of the falling raindrop to evaluate the below-cloud evaporation effect on isotopic composition, while method 2 evaluates its effect by directly measuring the variations of isotope composition.
In the revision, we did not take the isotopic method as a benchmark anymore. We just compared the two methods, and points out the flaws in each method.

After reading the full text with many times, we have changed the title to "A set of methods to evaluate the below-cloud evaporation effect on local precipitation isotopic composition: a case study in Xi'an, China." In the revised manuscript, we have reorganized the structure. Now, I think the main topics of this study have become more clear.

2. Due to the large number of assumptions needed in the below-cloud evaporation models, the sensitivity analysis has to be more prominent. Further, the different model simulations have to be introduced better. Currently, it is difficult to understand the difference between the models and the different simulations.

Thanks for your suggestion.

In the sensitivity test, we analyzed the effects of each input physical parameter, and compared their differences.

In addition, we have revised our expression in the section of " 2.5 Analytical methods". Here, in order to clearly introduce the two methods, we used method 1 and method 2 to represent them.

3. The manuscript is often repetitive. References to the methods are repeated in many places and the introduction to the sections and paragraphs are too general without leading towards the main topics of the paragraphs.

Thanks for your suggestion. We have shortened the introduction part, and moved some descriptions and information to the supplemental material as appendixes.

In fact, our manuscript consists of two parts: one is to use the $\Delta d\Delta\delta$-diagram to qualitatively identify the below-cloud processes in our site, and another is to compare two methods that are used to quantitatively evaluate the below-cloud effect on the local precipitation isotopic composition. In the revised manuscript, we have reorganized the structure. Now, I think the main topics of this study have become more clear.

**Specific comments:**
4. Introduction: this section is very long, consider shortening it.

Thanks for your suggestion. We have shortened the introduction part to 3 pages.

5. line 269: model -> do you mean mode? (same on line 281)

Yes, you are right. We have revised our expression in the revision.

6. line 269-271: For how long did you measure each liquid injection? Did you apply a drift correction?

Here, we chose high-precision mode to measure the liquid sample, and thus each injection needs about 10 minutes.

Yes, we applied drift correction. We added the information in the revision, and it reads "To correct the instrument drift, the three laboratory standards were repeatedly measured after measuring every 8 samples."

7. Line 287: "measured by a CTC Analytics autosampler...", do mean that the samples were injected using the autosampler? I expect the measurements to be done by the laser spectrometer.

Thanks for your suggestion. We have revised this sentence to "The standards are injected into the evaporator with a CTC Analytics autosampler, PAL HTC-xt (Leap Technologies, Carrboro, NC, USA), and measured by the laser spectrometer."

8. Lines 311-331: This paragraph seems out of place as it discusses a method that hasn't been introduced yet.

Yes, you are right, we have deleted this section in the revised manuscript.

9. Lines 334-335: The first part of this sentence is difficult to understand, consider reformulating.

Following your suggestion, we revised this sentence to "Since the water vapor concentration effect and isotopic composition dependency of the cavity ringdown spectrometer have been pointed out by many studies (e.g., Bastrikov et al., 2014; Benetti et al., 2014; Steen-Larsen et al., 2013; Weng et al., 2020), it is important to determine the isotopic composition-humidity correction response function."

10. Lines 344-347: For how long did you measure each liquid injection? Did you apply a drift correction?

Here, we chose high-precision mode to measure the liquid sample, and thus each injection needs about 10 minutes.

Yes, we applied drift correction. We added the information in the revision, and it reads "To correct the instrument drift, the three laboratory standards were repeatedly measured after measuring every 8 samples."

11. Lines 349-350: The humidity-isotope dependency as shown in S1a and S1b shows a dependency on the isotopic composition of the standards as reported by Weng et al. (2020). For example in S1b, LS-1 shows a decrease in $\Delta\delta2H$ with decreasing humidity while LS-3 shows an increase with decreasing humidity. The linear calibration function of $\delta2H$ does not take this into account. Therefore, the humidity-isotope calibration functions (eq 1 and 2) should be reconsidered to include this isotope-dependency.

Thanks for your suggestion. You are right that both the humidity and isotopic composition effects exist in our data. Therefore, we referred to Weng's method to recorrect our data. We have added the discussions in the supplemental material Appendix A.

By using the different correction methods, they show some differences in the most positive and negative end-members. In the revision, we adopted Weng's method to correct our data.

[Figure]

Figure 1 The relationships of $\delta^{18}O_v$ and $\delta^2H_v$ corrected by two different methods.

12. Line 355: "δhumidity calibration is the calibrated data for water vapor stable isotope" do you mean "... is the *humidity-dependency corrected* data.."?
Thanks for your suggestion, we have revised the expression to "where $\delta_{iso\text{-}hum\text{-}cor}$ is for isotopic composition-humidity dependency corrected water vapor isotopic composition at 20000 ppmv; $\delta_{meas}$ is the raw, measured isotopic composition at that humidity; h is the measured humidity; and a, b, and c are fitting coefficients for each water standard and isotope species. "

13. Line 266: representative -> do you mean "representativeness"?
We have revised the subtitle to "The representativeness of the data".

14. Line 373-374: "of which 100 precipitation samples have corresponding daily average water vapor isotopic results" Does this mean you compared the precipitation isotopes with daily average water vapour data? If yes, this would mean that you don't always compare the same time periods with each other. To compare preciptiation and water vapour, the water vapour isotopic composition should be averaged over the time period of the preciptiation event.
Following your suggestion, we have recalculated the per-event water vapor isotopic composition on each precipitation day, and compared them with the per-event precipitation isotopic composition. We revised the sentence to "In 2 years, a total of 514 days of water vapor isotopic composition measurements were carried out. For 141 precipitation samples, of which 100 precipitation samples have corresponding event-based water vapor isotopic results."

15. 388: "various" -> do you mean "different"?
The sentence has been revised to "Making use of stable water isotopes, Graf et al. (2019) introduced a ΔdΔδ-diagram to diagnose the below-cloud processes and their effects on vapor and precipitation isotopic composition, since equilibration and evaporation are two different processes and lead to different directions in the two-dimensional phase space of the ΔdΔδ-diagram."

16. Lines 389-397: δpv-eq and d-excesspv-eq are defined twice in different ways. On lines 390-391, it says that these variables represent the equilibrium vapour from the precipitation samples, on lines 396-397, it says that they represent the water vapour composition at cloud base.
Thanks for your suggestion. We have unified our statement in the revision. Now it reads "Here, the differences in the isotopic composition of precipitation-equilibrated vapor relative to the observed ground-level vapor can be expressed as:

$$\Delta\delta_v = \delta_{pv\text{-}eq} - \delta_{gr\text{-}v} \qquad\qquad \text{(eq. 2)}$$

$$\text{d-excess}_{pv\text{-}eq} - \text{d-excess}_{gr\text{-}v} \qquad\qquad \text{(eq. 3)}$$

where $\delta_{pv\text{-}eq}$ and $\delta_{gr\text{-}v}$ are the $\delta^2H$ ($\delta^{18}O$) of equilibrium vapor from precipitation and observed vapor near the ground, respectively, and d-excess$_{pv\text{-}eq}$ and d-excess$_{gr\text{-}v}$ are d-excess values of equilibrium vapor from precipitation and observed vapor near the ground, respectively."

17. Line 396: d-excesspv-eq and d-excess**pv-eq** -> d-excesspv-eq and d-excess**gr-v**

We have revised those expressions.

18. Sections 2.5.2 and 2.5.3: These two section introduce the two methods used to calculate the below-cloud effect. After reading these two section, I still didn't fully unterstand which assuptions were made. I think that part of the Appendix A has to be mentioned in 3.5.2 (e.g. how the isotopic composition of precipitation at the cloud base is estimated). A conceptual schematic of the properties (and assumptions) between cloud base and ground and how they differ between the two methods might help to understand the two models better.

Following your suggestion, we have seriously revised the discussions in this part. We have clearly pointed out the differences between the two methods, that is, "method 1 makes use of the mass change of the falling raindrop to evaluate the below-cloud evaporation effect on isotopic composition, while method 2 evaluates its effect by directly measuring the variations of isotope composition."

Please refer to "2.5.2 Below-cloud evaporation calculation: Method 1; 2.5.3 Below-cloud evaporation calculation: Method 2; Appendix C; and Appendix D for the detailed revision.

19. Section 2.5.3 The first and last paragraph repeat a lot of information already mentioned earlier in the manuscript.

Thanks for your suggestion, we have shortened the content, and deleted the repeated information.

20. Line 508-509: "the corresponding day's water vapour isotopic composition" why do you not compare the perevent mean water vapour isotopic composition? The water vapour isotopic composition changes strongly pre-, intra- and post-event (e.g. Aemisegger et al. 2015). If you average over the full day instead of the precipitation period, the average water vapour isotopic composition can differ strongly.

Yes, you are right. Following your suggestion, we have recalculated the per-event water vapor isotopic composition on each precipitation day, and compared them with the per-event precipitation isotopic composition.

It expressed like this "As shown in Fig. 2, the LMWL is: $\delta^2H_p=7.0\times\delta^{18}O_p+3.0$ based on event precipitation isotopic composition, and the local water vapor line (LWVL) is: $\delta^2H_v=7.8\times\delta^{18}O_v+15.1$ based on per-precipitation-event water vapor isotopic composition."

21. Line 513: Is there any seasonal cycle in the mean cloud base at your measurement location?

In the former edition, we assumed the cloud base is 1500m. Now, in the revision, we also used the empirical equation to define it:

$$Z=18400(1+\frac{T_{mean}}{273})\lg\frac{S_0}{S_{LCL}}$$

The calculated cloud base heights have large variations in each precipitation event (Fig. S5).

22. Line 525-526: "equilibrium prediction" -> do you mean "equilibrium water vapour from preciptiation"?

Yes, we have revised this sentence to "The reasonable agreement of observed and equilibrated water vapor isotopic composition has been reported by Jacob and Sonntag (1991), Welp et al. (2008), and Wen et al. (2010), however, they postulated the different relationships underlying the $\delta^{18}O_v$ and $\delta^{18}O_{pv\text{-}eq}$."

23. Line 534: How is $\delta18Ov\text{-}eq(1500m)$ calculated?

Here, we revised our statement, and it reads "we used the measured precipitation isotopic composition to deduce the water vapor isotopic composition at the ground level according to the liquid-vapor equilibrium isotope fractionation ($\delta^{18}O_{pv\text{-}eq}$), and compared it with observed water vapor ($\delta^{18}O_v$) in Fig. 3b."
For the $\delta^{18}O_v$ isotopic composition at the cloud base, please refer to supplemental material, Appendix D.

24. Lines 535-542: The sentence "our results indicate that it is possible to derive the isotope composition of atmospheric water vapor based on that of the precipitation in the semi-arid area." seems to contradict "It is worth noting that we do not propose to extract the water vapor isotope time series from precipitation data." What is your message here?

Thanks for your suggestion. We have rewritten the discussions, and now it reads "Although there is a good relationship between $\delta^{18}O_v$ and $\delta^{18}O_{pv\text{-}eq}$ in our data, the below-cloud evaporation has significant influence on the precipitation isotopic composition. Therefore, it should be cautious to derive the water vapor isotopic composition from the precipitation one."

25. Line 555: d-excess is 0 ‰ during equilibrium fractionation only at temperatures around 20°C.

Thanks for your suggestion. The sentence has been revised to "Traditionally, to qualitatively assess the below-cloud evaporation of raindrops, the value of d-excess$_p$ is a benchmark. Due to the differences in diffusivities of the individual water molecules in non-equilibrium fractionation, therefore, it will cause d-excess$_p$ to deviate from 0‰, which is a theoretical value under vapor-liquid equilibrium fractionation at temperatures around 20°C (Gat, 1996)."

26. Lines 560-561: Kinetic/non-equilibrium fractionation should be introduced earlier, and used consistently.

Thanks for your suggestion. The sentence has been revised to "Traditionally, to qualitatively assess the below-cloud evaporation of raindrops, the value of d-excess$_p$ is a benchmark. Due to the differences in diffusivities of the individual water molecules in non-equilibrium fractionation, therefore, it will cause d-excess$_p$ to deviate from 0‰, which is a theoretical value under vapor-liquid equilibrium fractionation at temperatures around 20°C (Gat, 1996)."
In addition, the expression of non-equilibrium fractionation has been unified in the manuscript.

27. Line 562-563: "Therefore, during the moisture transportation, the water vapor d-excess may be modified." What do you mean with *moisture*

*transportation*? Diffusion or large-scale advection? It is important to specify the scale of the process.

Thanks for your suggestion, we have revised the expression to "In addition, in the water molecules diffusion process, the water vapor d-excess$_v$ may be modified, and this enhances the uncertainty to gauge the below-cloud evaporation process by solely using d-excess$_p$."

28. Line 584-585: "...from through..." -> this is a repetition.

It has been revised to "However, as for snowfall event, it seems unreasonable to explain the strongly negative $\Delta\delta^2H_v$ by the raindrop size and rain rate (Fig. 4)."

29. Lines 591-593: Can you be more specific about what is the *new* learning from your results? Is has been known before, that snow does not interact strongly with sourrounding water vapour below the cloud base during its fall (e.g. Gedzelman and Arnold, 1994).

Thanks for your suggestion. The ΔdΔδ-diagram not only can be used to separate the below-cloud processes, but also to differ the precipitation types. Graf et al. (2019) only test the precipitation results on ΔdΔδ-diagram. Our snow sample supplements their data.

In order to specify our new learning, we revised the last sentence to "Our results suggest that in addition to raindrop size and rain rate, precipitation type is also an essential factor that influences the distribution of the data on the ΔdΔδ-diagram."

30. Line 603: △2Hv -> do you mean △δ2Hv? This notation isn't used consistently in the manuscript.

Yes, you are right. After checking the manuscript, the notations have been used consistently.

31. Line 603-621 and Fig.5: The discussed connection between meteorological conditions and the isotopic values is difficult to see in these figures. Boxplots instead of scatterplots might work better.

Thanks for your suggestion. We have deleted this part, instead, we discussed the meteorological controls in section 3.3.2: Meteorological controls on the two methods.

32. Line 615-617: I don't understand this sentence. Are you referring to the temperature dependency of equilibrium fractionation?

We have deleted the discussions.

33. Line 670: "below-unsaturation" -> what does that mean?

We have deleted that sentence.

**Figures:**
34. Fig. 1: The labels of the subfigures are very small
We have enlarged the labels of the subfigure in Figure 1.

[Figure]

35. Fig.4: The linear fit does not fit well to the data. Why do try to find a linear fit for snow and rain together? As these hydrometeors are influenced by different processes while falling, it is unlikely, that they lie on the same line in this diagram.
Following your suggestion, we have separately drawn the regression lines for snow and rain samples.

[Figure]

36. Fig.5: label text is very small. The dark red of very high values (e.g. temperature) is difficult to see.
We have deleted Figure 5.

---

## Referee Report (RR1)

Review acp-2022-576

Review of revised submission

**A set of methods to evaluate the below-cloud evaporation effect on local precipitation isotopic composition: a case study in Xi'an, China**

**Meng Xing, Weiguo Liu, Jing Hu, and Zheng Wang**

*Remark: In the following, I'm referring to the line numbers in the revised manuscript. The newly added text in the track change document differs partly from the text in the revised manuscript.*

**General:**

The revised manuscript by Xing et al. describes the qualitative and quantitative evaluation of below-cloud processes in a two year time series of isotopic composition in water vapour and precipitation at Xi'an. The authors considered many points addressed in the review (post-processing of water vapour measurements, sensitivity and uncertainty analysis, re-structuring of paper) which improve the scientific analysis and reader guidance of the paper.

Two main issues remain:

1. Even though the language quality has improved, I strongly recommend another language check by a native speaker before publication. There are many grammatical mistakes and sentences that are difficult to read which makes the scientific message difficult to understand.

2. On the discussion/comparison of the two methods:
   - Both methods use an important assumption which is that the surface water vapour is (moist) adiabatically connected to the cloud-base water vapour. In method 1, this assumption is used to calculate the cloud base height, temperature and pressure. In method 2, the isotopic composition of the cloud-base water vapour is calculated assuming a moist adiabatic ascent of the measured ground-level water vapour.
   This assumption only holds if the vertical column at the observation site is undisturbed by horizontal movement (as mentioned as assumption on lines 353-354). A discussion of the validity of these assumptions (under which circumstance can we assume a vertically undisturbed and connected column and where does the assumption not apply) is currently missing in the manuscript.

   - Further, it's important to mention more prominently that method 1 only includes below-cloud evaporation by construction while in method 2 other processes can still be included. I'm missing a discussion of these differences between the methods and the learnings from the comparison. This should be added as a discussion/conclusion point on which method should be applied under which conditions (i.e. under which meteorological conditions might an assumption of local (moist-) adiabatic (not be) valid) and how to possibly improve them in future studies (e.g. could the methods be improved to better represent below-cloud processes during snowfall?).

   Currently, some of these points are addressed at different locations throughout the manuscript (349-358, 536-538 (with respect to snowfall and low temperature), 667-674, Supplement C & D ). I recommend a summary of these points in the end.

**Specific:**

- Abstract:
    - The abstract needs a language check
    - Lines 33-37, the first sentence is difficult to read. I suggest to divide it into two sentences: "When hydrometeors fall from an in-cloud saturated environment towards the ground, especially in the arid and semi-arid regions, below-cloud processes may heavily alter the precipitation isotopic composition through equilibrium and non-equilibrium fractionation. If these below-cloud processes are not correctly identified, they can lead to misinterpretation of the precipitation isotopic signal. "
    - Line 60, "therefore" in last sentence: It's not clear to me how this sentence connects to the previous sentence(s).

- Lines 99-101: "The equilibrium fractionation would not change the d-excess while the non-equilibrium diffusional process would result in a decrease of d-excess in rain (Fisher, 1991; Merlivat and Jouzel, 1979)"

    This is not correct as a general statement. D-excess can change during equilibrium fractionation, depending on ambient temperature. Possibly rephrase to "Equilibrium fractionation does not substantially change d-excess while …"

- Line 128: what do you mean with "initial signal"? The cloud-base signal?

- Line 205: "after filtration": do you mean: "second the samples were filtered, and then immediately..."

- Lines 459-477: this paragraph is difficult to follow, many sentences are difficult to understand due to poor languange;  e.g. "Hence, rain/snow formed under such circumstances, their isotopic signals will be less impacted by the environmental factors during its falling."
    Please, check the language in this paragraph.

- Fig.4: It's nice that you added two regression lines and interesting that the linear regression for rain is similar to the results by Graf et al. (2019).
    The colors of the regression lines don't match the colors of the markers for the respective precipitation type in the new figure. This makes the figure difficult to read.

- Lines 529-532:
    "During the supersaturation process, vapor deposition occurs over ice (Jouzel and Merlivat, 1984), which may cause the snow isotopic composition at the ground to be more depleted than its formation height."

    It is not entirely clear to me where this vapor deposition takes place: is it vapor deposition during falling or in the cloud before the snowfall? Why does vapour deposition during snow fall decrease $\delta^{18}O$ and $\delta D$ of the snow? Whether the isotopic composition of snow becomes less depleted in heavy isotopes during this process depends on the isotopic composition of the water vapour below the cloud relative to in-cloud vapour.

- Lines 534-536: "The diameter of the raindrop used to determine the terminal velocity and evaporation intensity (Supplemental material, eq. 10-13) does not take into account the snowfall factor which results in great uncertainty in method 1.

    What's "the snowfall factor"? Do you mean the different relationship of fall velocity to hydrometeor size for snow flakes and rain drops?

- Lines 551-552: "The significant difference in winter might be related to the supersaturation process."

So far, you've mentioned supersaturation and vapour deposition as a possible mechanism leading to negative ΔδD during winter. Instead of referring to this process, I'd refer to the presence of solid precipitation during winter time. E.g.: "The significant difference in winter might be related to the predominance of solid precipitation which is not accounted for in method 1."

- Lines 562-564: "Wang et al. (2016b) explicitly pointed out that among the parameters of temperature, precipitation amount, RH, and raindrop diameter, RH generally plays a decisive role on Δd-excess in the below-cloud evaporation process."

  You're only showing ΔδD, how about the role of these parameters for Δdexcess in your data? This might be too much to add in this manuscript but mentioning the results from Wang et al. (2016b) on Δdexcess rises this question.

- Section 3.4: This part seems out of place and I didn't learn anything new while reading it (especially after seeing Figure S4 in Section 3.3.1). Further, it seems partly a repetition of the lines 549-552. Is this section needed?

- Fig.7: out of curiosity: there seems to be a seasonality on the effect of a temperature decrease on ΔδD, which is opposite for method 1 and 2. How do you explain the (opposite) seasonality?

- Lines 686-687: "the precipitation and water vapor isotopic compositions have a good relationship"

  What do you mean with "good relationship"?

---

## Author Response (AR2)

Reviewer #1

**General comments:**

1. Even though the language quality has improved, I strongly recommend another language check by a native speaker before publication. There are many grammatical mistakes and sentences that are difficult to read which makes the scientific message difficult to understand.

Thanks for your suggestion, we have used the professional language editing company, American Journal Experts (AJE), to help us to polish our manuscript. Now, we believe our language quality has reached the requirements of the journal.

2. Both methods use an important assumption which is that the surface water vapour is (moist) adiabatically connected to the cloud-base water vapour. In method 1, this assumption is used to calculate the cloud base height, temperature and pressure. In method 2, the isotopic composition of the cloud-base water vapour is calculated assuming a moist adiabatic ascent of the measured ground-level water vapour.

This assumption only holds if the vertical column at the observation site is undisturbed by horizontal movement (as mentioned as assumption on lines 353-354). A discussion of the validity of these assumptions (under which circumstance can we assume a vertically undisturbed and connected column and where does the assumption not apply) is currently missing in the manuscript.

Thanks for your suggestion. Yes, you are right, these assumptions are very important, and should be clearly pointed out in the manuscript. Now, we added this information in Section 2.5, and it reads "Here, it should be noted that both methods use an important assumption which is that the surface water vapor is (moist) adiabatically connected to the cloud-base water vapor in the air column. In method 1, this assumption is used to calculate the cloud base height, temperature, and pressure (Appendix C, Eq. 14-16). In method 2, the isotopic composition of the cloud-base water vapor is calculated assuming a moist adiabatic ascent of the measured ground-level water vapor (Appendix C, Eq. 22). In addition, in method 2, we assumed that the raindrop isotopic composition ($\delta_{cb-p}$) at the cloud base is in equilibrium with the surrounding water vapor, and the observed ground-level precipitation isotopic composition ($\delta_{gr-p}$) includes the processes of evaporation, growth, and isotopic equilibrium with the surrounding vapor. Furthermore, the air column is assumed that there is no horizontal advection into or out of it, and no updraft or downdraft of the air masses during the hydrometeors' falling. That means the vertical column at the observation site is undisturbed by horizontal movement. These assumptions only hold if a single vertical column extends from the ground to the cloud-base height. When the rain events during which the single column is affected by the surrounding air, these assumptions become invalid."

3. Further, it's important to mention more prominently that method 1 only includes below-cloud evaporation by construction while in method 2 other processes can still be included. I'm missing a discussion of these differences between the methods and the learnings from the comparison. This should be added as a discussion/conclusion point on which method should be applied under which conditions (i.e. under which meteorological conditions might an

assumption of local (moist-) adiabatic (not be) valid) and how to possibly improve them in future studies (e.g. could the methods be improved to better represent below-cloud processes during snowfall?).

Thanks for your suggestion, we have added the content to the conclusions. "4. Considering the assumption that the surface water vapor is (moist) adiabatically connected to the cloud-base water vapor, therefore, the validation of the two methods is for frontal precipitation or convective precipitation. Here, method 1 only includes below-cloud evaporation by construction while in method 2 other processes can still be included, such as supersaturation. Therefore, both methods are suited to study the below-cloud evaporation effect (no statistical differences in $\Delta\delta^2H_p$ for rainfall events), however, if other below-cloud processes are included, applying method 2 is the better choice. In future studies, further high-resolution observations of vertical profiles of precipitation and water vapor isotopes, whether tower-based or aircraft-based, have the potential to greatly improve constraints on below-cloud processes."

**Specific comments:**

4. The abstract needs a language check

We have carefully revised the abstract part, and now it reads "When hydrometeors fall from an in-cloud saturated environment toward the ground, especially in arid and semiarid regions, below-cloud processes may heavily alter the isotopic composition of precipitation through equilibrium and non-equilibrium fractionations. If these below-cloud processes are not correctly identified, they can lead to misinterpretation of the precipitation isotopic signal. To correctly understand the environmental information recorded in the precipitation isotopes, qualitatively analyzing the below-cloud processes and quantitatively calculating the below-cloud evaporation effect are two important steps. Here, based on two years of synchronous observations of precipitation and water vapor isotopes in Xi'an, we compiled a set of effective methods to systematically evaluate the below-cloud evaporation effect on local precipitation isotopic composition. The $\Delta d\Delta\delta$-diagram is a tool to effectively diagnose below-cloud processes, such as equilibration or evaporation, because the isotopic differences ($\delta^2H$, d-excess) between the precipitation-equilibrated vapor and the observed vapor show different pathways. By using the $\Delta d\Delta\delta$-diagram, our data show that evaporation is the major below-cloud process in Xi'an, while snowfall samples retain the initial cloud signal because they are less impacted by the isotopic exchange between vapor and solid phases. Then, we chose two methods to quantitatively characterize the influence of below-cloud evaporation on local precipitation isotopic composition: one is based on the raindrop's mass change during its falling (hereafter referred to as method 1); the other is dependent on the variations in precipitation isotopic composition from the cloud base to the ground (hereafter referred to as method 2). By comparison, we found that there are no significant differences between the two methods in evaluating the evaporation effect on $\delta^2H_p$, except for snowfall events. The slope of evaporation proportion to the variation in $\delta^2H$ ($F_i/\Delta\delta^2H$) is slightly larger in method 1 (1.0 ‰/%) than in method 2 (0.9 ‰/%). Additionally, both methods indicate that the evaporation effect is weak in autumn and heavy in spring. Through a sensitivity test, we found that in two methods, relative humidity is the most sensitive parameter, while the

temperature shows different effects on the two methods. Therefore, we concluded that both methods are suited to investigate the below-cloud evaporation effect, while in method 2, other below-cloud processes, such as supersaturation, can still be included. By applying method 2, the diagnosis of below-cloud processes and the understanding of their effects on the precipitation isotopic composition will be improved."

5. Lines 33-37, the first sentence is difficult to read. I suggest to divide it into two sentences: "When hydrometeors fall from an in-cloud saturated environment towards the ground, especially in the arid and semi-arid regions, below-cloud processes may heavily alter the precipitation isotopic composition through equilibrium and non-equilibrium fractionation. If these below-cloud processes are not correctly identified, they can lead to misinterpretation of the precipitation isotopic signal. "

Thanks for your suggestion, we have divided this sentence into two sentences, and now it reads "When hydrometeors fall from an in-cloud saturated environment towards the ground, especially in the arid and semi-arid regions, below-cloud processes may heavily alter the precipitation isotopic composition through equilibrium and non-equilibrium fractionation. If these below-cloud processes are not correctly identified, they can lead to misinterpretation of the precipitation isotopic signal. "

6. Line 60, "therefore" in last sentence: It's not clear to me how this sentence connects to the previous sentence(s).

We have revised the last sentence, now it reads "Therefore, we concluded that both methods are suited to investigate the below-cloud evaporation effect, while in method 2 other below-cloud processes, such as supersaturation, can still be included. By applying method 2, the diagnosis of below-cloud processes and the understanding of their effects on the precipitation isotopic composition will be improved."

7. Lines 99-101: "The equilibrium fractionation would not change the d-excess while the non-equilibrium diffusional process would result in a decrease of d-excess in rain (Fisher, 1991; Merlivat and Jouzel,1979)"

This is not correct as a general statement. D-excess can change during equilibrium fractionation, depending on ambient temperature. Possibly rephrase to "Equilibrium fractionation does not substantially change d-excess while …"

Thanks for your suggestion, we have rephrased the sentence to "Equilibrium fractionation does not substantially change d-excess, while non-equilibrium diffusional process would result in a decrease of d-excess in rain (Fisher, 1991; Merlivat and Jouzel, 1979)."

8. Line 128: what do you mean with "initial signal"? The cloud-base signal?

Yes, you are right, we have changed the "initial signal" into the "cloud-base signal".

9. Line 205: "after filtration": do you mean: "second the samples were filtered, and then immediately..."

Yes, you are right. Following your suggestion, the sentence now reads "The

snowfall samples were first melted at room temperature in closed plastic bags, second the samples were filtered, and then immediately poured into 100 ml polyethylene bottles."

10. Lines 459-477: this paragraph is difficult to follow, many sentences are difficult to understand due to poor language; e.g. "Hence, rain/snow formed under such circumstances, their isotopic signals will be less impacted by the environmental factors during its falling."
Please, check the language in this paragraph.
Thanks for your suggestion, we have rephrased this paragraph, and it reads "Based on the results from numerical simulations and in situ observations, Graf et al. (2019) concluded that raindrop size and precipitation intensity are two important factors for determining below-cloud processes. For example, precipitation with large raindrops and heavy intensities is less affected by below-cloud processes because of the shorter residence time of raindrops in the atmospheric column with a faster fall velocity. Therefore, they are less affected by the evaporation and equilibration processes on their falling way toward the ground surface, and the $\Delta\delta^2H_v$ is more negative. It is worth noting that in the case of not considering the factors of raindrop size and rain rate, the different precipitation types also show a clear distribution on the $\Delta d\Delta\delta$-diagram, as almost all the snowfall samples have negative $\Delta\delta^2H_v$ values (Fig. 4). Theoretically, snowfall events normally occur in low-temperature conditions and correspond to weak evaporation. Furthermore, the diffusion speed of the ice phase (solid) to vapor is lower than that of liquid to vapor. Hence, under such conditions, the isotopic signals of rain/snow are less affected by the below-cloud processes during falling. This leads $\Delta\delta$ to be more negative with decreasing temperature, such as the observed phenomenon in the post-frontal precipitation isotopes in Graf et al.'s (2019) study. Additionally, on the $\Delta d\Delta\delta$-diagram, the snow samples with positive $\Delta d$-excess$_v$ (in the second quadrant) may be related to the supersaturation process, as the liquid has unusually high d-excess$_p$ for the non-equilibrium fractionation of supersaturation (Deshpande et al., 2013; Jouzel and Merlivat, 1984). We conclude that in addition to raindrop size and rain rate, precipitation type is also an essential factor in determining the data distributions on the $\Delta d\Delta\delta$-diagram."

11. Fig.4: It's nice that you added two regression lines and interesting that the linear regression for rain is similar to the results by Graf et al. (2019).
The colors of the regression lines don't match the colors of the markers for the respective precipitation type in the new figure. This makes the figure difficult to read.

Thanks for your suggestion. In order to make the colors of the regression lines match the colors of the markers, we have changed the colors in Fig.4.

[Figure]

12. Lines 529-532:
"During the supersaturation process, vapor deposition occurs over ice (Jouzel and Merlivat, 1984), which may cause the snow isotopic composition at the ground to be more depleted than its formation height."
It is not entirely clear to me where this vapor deposition takes place: is it vapor deposition during falling or in the cloud before the snowfall? Why does vapour deposition during snow fall decrease δ18O and δD of the snow? Whether the isotopic composition of snow becomes less depleted in heavy isotopes during this process depends on the isotopic composition of the water vapour below the cloud relative to in-cloud vapour.
That is a good question. Here, the vapor deposition takes place in the cloud before the snowfall. We cited the hydrometeors formation mechanism in Graf's et al. (2019) study (Appendix A: A2.1 Growth by vapor deposition).
Precipitation formation in both mixed-phase and ice clouds occurs by deposition of vapor on ice particles. Non-equilibrium fractionation due to supersaturation with respect to ice is taken into account with a kinetic fractionation factor $\alpha_k$.

$$R_{cb\text{-}ps}= \alpha_s\alpha_k R_{cb\text{-}v} \qquad \text{Eq.1}$$

where $R_{cb\text{-}ps}$ is the isotopic composition of precipitation at the cloud base, $\alpha_s$ is the equilibrium fractionation coefficient with respect to the solid phase, $R_{cb\text{-}v}$ is the isotopic composition of water vapor at the cloud base, and $\alpha_k$ is the kinetic fractionation factor that vapor deposition occurs over ice during the supersaturation process, which can be written in terms of properties of the bulk gas:

$$\alpha_k=\frac{S_i}{\alpha_s D/D'(S_i-1)+1} \qquad \text{Eq.2}$$

where $S_i$ is the supersaturation over ice, and D/D' is the ratio of the diffusion coefficients of the light and heavy isotopes.

In the equilibrium state, the isotopic fractionation between the solid and vapor phases follows a temperature-dependent factor:

$$R_{cb\text{-}pe}= \alpha_s R_{cb\text{-}v} \qquad \text{Eq.3}$$

Here, the kinetic fractionation factor $\alpha_k$ is not taken into account.

Because the D/D' of HDO or $H_2^{18}O$ is lower than 1, $\alpha_k$ is lower than the unity. Hence, the $R_{cb-ps}$ calculated by Eq.1 is smaller than the $R_{cb-pe}$. The $R_{cb-ps}$ and $R_{cb-pe}$ correspond to $\delta_{gr-p}$ and $\delta_{cb-p}$ in Eq. 4, respectively.

$$\Delta\delta_p = \delta_{gr-p} - \delta_{cb-p} \qquad\qquad Eq.4$$

Therefore, during the supersaturation process, the snow isotopic composition observed at the ground is more depleted than its formation height.

Here, to make our expression more clear, we revise the sentence to "During the supersaturation process, vapor deposition takes place over ice in the cloud (Jouzel and Merlivat, 1984) with non-equilibrium fractionation (the kinetic fractionation factor $\alpha_k<1$), leads the effective isotopic fractionation factor ($\alpha_{eff} = \alpha_{eq}\alpha_k$) to be smaller than the equilibrium fractionation coefficient ($\alpha_{eq}$), and results in the ground observed $\delta_{gr-p}$ of solid precipitation (snow) more depleted than the calculated $\delta_{cb-p}$ under equilibrium fractionation (in Eq. 7).

13. Lines 534-536: "The diameter of the raindrop used to determine the terminal velocity and evaporation intensity (Supplemental material, eq. 10-13) does not take into account the snowfall factor which results in great uncertainty in method 1.
What's "the snowfall factor"? Do you mean the different relationship of fall velocity to hydrometeor size for snow flakes and rain drops?
Yes, you are right. Following your suggestion, the sentence now reads "The diameter of the raindrop used to determine the terminal velocity and evaporation intensity (Supplemental material, Eq. 10-13) does not take into account the different relationship of fall velocity to hydrometeor size for snowflakes and raindrops, which results in great uncertainty in method 1."

14. Lines 551-552: "The significant difference in winter might be related to the supersaturation process."
So far, you've mentioned supersaturation and vapour deposition as a possible mechanism leading to negative $\Delta\delta D$ during winter. Instead of referring to this process, I'd refer to the presence of solid precipitation during winter time. E.g.: "The significant difference in winter might be related to the predominance of solid precipitation which is not accounted for in method 1."
Thanks for your suggestion, we have revised this sentence to "The significant difference in winter might be related to the predominance of solid precipitation which is not accounted for in method 1."

15. Lines 562-564: "Wang et al. (2016b) explicitly pointed out that among the parameters of temperature, precipitation amount, RH, and raindrop diameter, RH generally plays a decisive role on $\Delta$d-excess in the below-cloud evaporation process."
You're only showing $\Delta\delta D$, how about the role of these parameters for $\Delta$dexcess in your data? This might be too much to add in this manuscript but mentioning the results from Wang et al. (2016b) on $\Delta$dexcess rises this question.
You are right. According to your suggestion, we have deleted the cited reference.

16. Section 3.4: This part seems out of place and I didn't learn anything new while reading it (especially after seeing Figure S4 in Section 3.3.1). Further, it seems partly a repetition of the lines 549-552. Is this section needed?

Following your suggestion, we have deleted this paragraph.

17. Fig.7: out of curiosity: there seems to be a seasonality on the effect of a temperature decrease on $\Delta\delta D$, which is opposite for method 1 and 2. How do you explain the (opposite) seasonality?

Yes, you are right. Here, the method 1 and 2 show a weak seasonality on the effect of a temperature decrease on $\Delta\delta D$, but their trends are opposite.

In method 1, the temperature is related to the calculation of $F_r$, $\alpha$, $\gamma$, and $\beta$ (Eq. 5), and thus the complex calculating processes offset its impact on the $\Delta\delta D$, which results in less sensitivity of temperature on the variations of $\Delta\delta D$.

By comparison, the temperature highly impacts the calculation of $\delta_{cb-p}$ in method 2 (Eq. 5). In addition, we used the different equations to calculate the equilibrium factors when the temperature is below or above 0°C. For example, when the temperature is greater than 0 °C, we use the equation of Horita and Wesolowski (1994) to calculate $^2\alpha$ and $^{18}\alpha$, when the temperature is below 0 °C, the equilibrium fractionation factor proposed by Ellehoj et al. (2013) is used (supplemental material: Appendix B). Therefore, the different equations may cause the seasonality in method 2.

18. Lines 686-687: "the precipitation and water vapor isotopic compositions have a good relationship" What do you mean with "good relationship"?

Thanks for your suggestion, we have changed the sentence into "In arid areas, the precipitation and water vapor isotopic compositions are closely related, and therefore the joint observation of the two tracers could provide more information on the precipitation processes."

Reviewer #2

**General technical issues:**

1. "eq." should be written in capital letter throughout the manuscript and supplement (thus "Eq.", see ACP guidelines)

Thanks for your suggestion, we have changed "eq." into "Eq." throughout the manuscript and supplement.

2. There should be a colon or full stop after Figure [Number], thus e.g "Figure 1:" or "Figure 1." instead of "Figure 1"

Thanks for your suggestion, we have added a full stop after Figure [Number] throughout the manuscript and supplement.

**Specific comments:**

3. P1, L60ff: It is still not clear why the two methods improve our understanding. Please clearly state what is improved. What do we derive from these methods? Which method is better? Are both methods well suited to investigate below cloud processes or are there some restrictions for one or the other method? This should be clearly stated in the abstract.

Thanks for your suggestion. To more clearly express our intention, we have revised the last sentence to "Therefore, we concluded that both methods are suited to investigate the below-cloud evaporation effect, while in method 2 other below-cloud processes, such as supersaturation, can still be included. By applying method 2, the diagnosis of below-cloud processes and the understanding of their effects on the precipitation isotopic composition will be improved."

4. P4, L128: What do you mean with "initial signal"? Is this found in the delta diagram?

Thanks for your suggestion, we have changed the "initial signal" into the "cloud-base signal"

5. P9, L272: Why does this function need to be determined? Why is this correction necessary?

Many studies have pointed out that the water vapor isotopic composition measured by cavity ringdown spectrometer has the humidity (water vapor mixing ratio) dependency, especially at the low water vapor mixing ratios. Furthermore, Weng et al., (2020) reported that the isotope composition of water vapor has a substantial and systematic impact on the mixing ratio dependency. If you want to get the accurate water vapor isotopic result, you need to build the relationship between the water vapor isotopic composition and its mixing ratio. Therefore, it is necessary to determine the isotopic composition-humidity correction response function.

To express more clearly, we have revised the sentence to "The water vapor concentration effect and isotopic composition dependency of the cavity ringdown spectrometer have been pointed out by many studies (e.g., Bastrikov et al., 2014; Benetti et al., 2014; Steen-Larsen et al., 2013; Weng et al., 2020). In order to minimize the uncertainty from the measurement, it is important to

determine the isotopic composition-humidity correction response function."

6. P11, L354: What is meant here with "updraft" and "downdraft"?
Here, we want to express that the horizontal and vertical air motion are neglected, while the updraft or downdraft corresponds to the vertical motion of air masses.
We have revised this sentence, and now it reads "In addition, during the hydrometeors falling we assumed that there is no horizontal advection into or out of the column, and no updraft or downdraft of the air masses."

7. P11, L366: What is SPSS? What does this p value mean? Why has this value been chosen. This should be clarified in the manuscript.
SPSS is the abbreviation of Statistical Package for Social Sciences. p represents the level of confidence.
Following your suggestion, we have revised this sentence to "To compare the difference between the two methods, the independent t-test was performed on Statistical Package for Social Sciences (SPSS 13.0, Inc., Chicago, US), followed by setting the significant statistical difference at the p=0.05 level of confidence."

8. P13, L423-424: Sentence is not clear. It may be that there is just a "the" missing, but what exactly is meant with two monthly equilibrated water vapour values? Are this values observed on a two monthly basis or derived (averaged?) over two years?
Thanks for your suggestion, we have revised this sentence to "Jacob and Sonntag (1991) suggested that the water vapor isotopic composition is possible to be deduced from the corresponding precipitation isotopic composition, but Wen et al. (2010) speculated that the equilibrium method cannot accurately predict the ground-level water vapor isotopic composition in arid and semiarid climates because of two the monthly equilibrated water vapor values in April and November deviating from the observed values."

9. P19, L581ff: Clearly write what the differences are. To make these more clearly I would suggest to write that the same input parameters as for method 1 have been used except precipitation amount.
Following your suggestion, we have revised the sentence to "In method 1, the input physical parameters include temperature, RH, surface pressure, and precipitation amount. In method 2, the same input parameters as for method 1 have been used except for precipitation amount."

10. P21, L646: Be more precise. Have been added quadratically to what?
Thanks for your suggestion. To make the sentence more clearly, we have revised it to "Hence, the lower and upper limits of the above used input parameters in for method 1 and method 2 are used to quantify the uncertainties and add them quadratically to ascertain the total uncertainty (Rangarajan et al., 2017; Wu et al., 2022)."

11. P21, L652: Change the order of the sentence parts so that it reads: Before exploring……it is important…. since ……….

Thanks for your suggestion. Now, it reads "Before exploring the information contained in the precipitation isotopes, it is important to clearly know the variation of precipitation isotopic composition during its falling, since the below-cloud evaporation is very common in arid and semi-arid regions."

12. P23, L707ff: As in the abstract, also here in the conclusion a clear message should be provided. When should one use method 1 and when method 2? Which of the two methods is better or are the same result derived?

Thanks for your suggestion, we have added a clear statement in the conclusion. Now, it reads "4. Considering the assumption that the surface water vapor is (moist) adiabatically connected to the cloud-base water vapor, therefore, the validation of the two methods is for frontal precipitation or convective precipitation. Here, method 1 only includes below-cloud evaporation by construction while in method 2 other processes can still be included, such as supersaturation. Therefore, both methods are suited to study the below-cloud evaporation effect (no statistical differences in $\Delta\delta^2H_p$ for rainfall events), however, if other below-cloud processes are included, applying method 2 is the better choice. In future studies, further high-resolution observations of vertical profiles of precipitation and water vapor isotopes, whether tower-based or aircraft-based, have the potential to greatly improve constraints on below-cloud processes."

**Technical corrections:**
13. Title: in Xi´an -> for Xi´an

Have done.

14. P2, L40: delete "a" rephrase as follows: …based on two-years of synchronous observations of precipitation……

Have done.

15. P2, L46: add "the" -> By using the

Have done.

16. P2, L58: add "we found that" -> Through the sensitivity test we found that relative humidity ……

Have done.

17. P2, L60: "following" is not the right term here. I would suggest to rephrase as follows: Therefore, by applying the two methods, the diagnosis of below-cloud processes and the understanding of their effects on the precipitation isotopic composition can be improved."

Have done.

18. P4, L105: rephrase as follows: "…..and a slope lower than 8.0 points to a non-equilibrium fractionation, such……." Further, I am not sure if it rather should non-equilibrium fractionation process.

Thanks for your suggestion, we have changed the sentence to "Generally, the LMWL's slope is approximately equal to 8.0 belonging to equilibrium fractionation, and a slope deviating from 8.0 is related to a non-equilibrium fractionation, such as the re-evaporation of raindrops.

19. P4, L120 and 122: add "the" before delta-diagram (twice) and LMWL (once).
Have done.

20. P4, L125: works -> work
Have done.

21. P4, L139: frame is not the correct term here. Rather "region" or "area". What is meant here with "simple"?
We have revised the sentence to "Froehlich et al. (2008) adapted the Stewart model and then assessed the change in d-excess due to below-cloud evaporation in the European Alps."

22. P5, L155: Change sentence to: However, so far these have not been systematically compared.
Have done.

23. P6, L178: remove "year"
Have done.

24. P7, L206: delete "a"
Have done.

25. P7, 207: Since the measurements are performed based on samples I would rather write "analysed" than "measured".
Have done.

26. P7, L216: same here
Have done.

27. P7, L224 and 225: Spaces are missing. You could also write VSMOW-GISP (Vienna Standard Mean Ocean Water – Greenland Ice Sheet Precipitation).
Have done.

28. P7, L227: This sentence is not clear and I would suggest to rephrase as follows: To correct the instrument drift, the instrument was repeatedly calibrated with the laboratory standards after analysing 8 samples.
Have done.

29. P9, L298: lead to -> encounter
Have done.

30. P9, L300: a -> the
Have done.

31. P10, L310: add "the" -> of the equilibrium
Have done.

32. P10, L330: variations -> variation
Have done.

33. P11, L341: isotope -> isotopic
Have done.

34. P11, L344: is able to calculate -> can be calculated
Have done.

35. P11, L349: Be more precise. Which method? Method 1 or method 2?
Have done.

36. P11, L352: isotopically -> isotopic
Have done.

37. P11, L357: more works need to -> more work is needed to
Have done.

38. P11, L358: delete "the"
Have done.

39. P11, L375: the LWML is -> the LMWL is defined as……. (or can be calculated by…….)
Have done.

40. P11, L376: same here for LWVL
Have done.

41. P11, L376: add "the" -> based on the per-precipitation-event water vapour
Have done.

42. P12, L378: of LMWL -> of the LWML
Have done.

43. P12, L378: are 8.0 and 10.0…… -> have a slope of 8.0 and 10.0….
Have done.

44. P12, L382: little -> somewhat
Have done.

45. P12, L382: of LMWL -> of the LMWL
Have done.

46. P12, 383: may also relate to -> may also be related to
Have done.

47. P12, L388:which -> where; Further, I would suggest to write "with the former being generally more negative when the latter"
Have done.

48. P12, L391: add "being" -> composition being more positive
Have done.

49. P12, L397: add "the" -> the precipitation
Have done.

50. P13, L400: add "the" -> with the observed
Have done.

51. P13, Figure 3 caption: dash-dot -> dash-dotted
Have done.

52. P13, L409: Add in which figure.
Have done.

53. P13, L413: deviation -> deviate
Have done.

54. P14, L429: clearly state which one? From the isotope composition?
Have done.

55. P14, L444: richer -> more
Have done.

56. P15, L466: event happens -> events happen
Have done.

57. P15, L467: corresponds -> correspond
Have done.

58. P15, L492: move "the" behind "in"
Have done.

59. P16, L493: scope is not the correct term here. It should rather read "area of" or "region of".
Have done.

60. P16, L496: works -> work
Have done.

61. P16, L500: "rich" is not the correct term. Use "valuable".
Have done.

62. P16, L506: The section header should rather read "Quantitative evaluation of the below-cloud evaporation effect derived from the two methods".
Have done.

63. P16, L508, L509 and L510: range -> ranges
Have done.

64. P16, L513: delete "a"
Have done.

65. P16, L514 and L515: in -> for or better write "derived from"
Have done.

66. P16, L514 and L515: delete "+-" before standard deviation
Have done.

67. P17, L523: show -> appears
Have done.

68. P17, L528: could not -> can not or better write "is always positive" instead of "be a negative number".
Have done.

69. P17, L538: when the -> for
Have done.

70. P19, L583: test -> tests
Have done.

71. P19, L591: at -> to
Have done.

72. P19, L597: have positive impact -> have a positive impact
Have done.

73. P20, L629: deciding -> determining
Have done.

74. P21, L643: section 2.4 -> Sect. 2.4
Have done.

75. P21, L645: add "used" -> above used input parameters
Have done.

76. P21, L645: in -> for
Have done.
77. P22, L691: add "the" -> of the local RH
Have done.

78. P23, L692: validates -> evaluates (?)
Have done.

79. P23, L705. difference -> differences
Have done.

80. P23, L707: add "we found that" after sensitivity analysis
Have done.

**Supplement:**
81. General: Add page numbers
Have done.

82. eq. -> Eq.
Have done.

**Specific comments:**
83. sentence after Eq. 14: add "the" -> the average
Have done.

84. P6: What is Q1 and Q2?
Q1 and Q2 are two parameters that are used to calculate the evaporation intensity of the falling drops, and there are no specific expressions for these two parameters. The values of Q1 and Q2 for specific conditions, i.e., T=0°C, 10°C, 20°C, 30°C, 40°C; D=0.01cm, 0.02cm, 0.03cm, ... , 0.44 cm; and h=10%, 20%, 30%, ..., 100%, were presented by Kinzer and Gunn (1951).
According to previous research, Wang et al., (2021) gave two approximative formulas to respectively calculate Q1 and Q2:

$$Q1=(-0.2445T+131.28)(0.1D)^{1.6139}$$
$$Q2=(-0.73h+0.7264)e^{(-0.002h+0.0371)T}$$

Based on Kinzer and Gunn (1951), the Pearson's determination coefficient ($R^2$) between the observations and estimates is 0.9826 for Q1 and 0.9942 for Q2 ($p < 0.0001$).

85. Appendix D, second paragraph: cloud base follows -> cloud base that follows vertical distribution of what?

Thanks for your suggestion, we have revised this sentence, and it reads "Therefore, the water vapor isotopic composition at the cloud base that follows the vertical distribution of Rayleigh distillation can be described by the following equation (Araguás-Araguás et al., 2000; Deshpande et al., 2010)"

86. Paragraph after Eq. 22: add "the" twice -> denotes the scale height of the atmospheric water vapour

Have done.

87. can use eq. 14 to calculate -> can be calculated by using Eq. 14

Have done.

88. Appendix E: in the two -> by the two

Have done.

89. defined -> derived (?)

Have done.

Wang, S., Jiao, R., Zhang, M., Crawford, J., Hughes, C.E., Chen, F., 2021. Changes in Below-Cloud Evaporation Affect Precipitation Isotopes During Five Decades of Warming Across China. J. Geophys. Res. Atmos. 126, 1–17. https://doi.org/10.1029/2020JD033075

Weng, Y., Touzeau, A., Sodemann, H., 2020. Correcting the impact of the isotope composition on the mixing ratio dependency of water vapour isotope measurements with cavity ring-down spectrometers. Atmos. Meas. Tech. 13, 3167–3190. https://doi.org/10.5194/amt-13-3167-2020

---

## Author Response (AR3)

Reviewer #2

1. p9, Eq.1: Adding "$\Delta\delta corr=$" (or similar) in front of the equation would make it easier to understand that the equation represents the correction term.

Thanks for your suggestion, we have added the "$\Delta\delta_{corr}=$" in front of equation 1.

2. p11, 359-361: "When the rain events during which the single column is affected by the surrounding air, these assumptions become invalid." Do you mean: "If the vertical column is affected by lateral entrainment of surrounding air, these assumptions become invalid. " ?

Yes, you are right. Following your suggestion, we have revised this sentence to "If the vertical column is affected by lateral entrainment of surrounding air, these assumptions become invalid."

3. p15, 472: "ground surface" -> use either "ground" or "surface".

Thanks for your suggestion, we have revised this sentence to "Therefore, they are less affected by the evaporation and equilibration processes on their falling way toward the ground"

4. p15, 479: instead of "rain/snow" -> "hydrometeor"

Thanks for your suggestion, we have changed "rain/snow" into "hydrometeor".

5. p15, 483: instead of "liquid" -> "water"

Thanks for your suggestion, we have changed "liquid" into "water".

6. p23, 700-702: "Considering the assumption that the surface water vapor is (moist) adiabatically connected to the cloud-base water vapor, the validation of the two methods is for frontal precipitation or convective precipitation."

I don't agree that a vertical connection between the cloud base and the ground is generally present for frontal precipitation, especially for precipitation along a warm front. Due to the forward-slanted vertical orientation of a warm front, upon arrival of the warm front, precipitation can fall into the prefrontal air that is not connected to moisture in the cloud. Due to the importance of lateral movement during frontal precipitation, it is possible that surface and cloud-based moisture are not connected.

Yes, you are right. After considering the weather condition you mentioned, we have revised the sentence to "Considering the assumption that the surface water vapor is (moist) adiabatically connected to the cloud-base water vapor, the validation of the two methods is for specific weather conditions, such as convective precipitation."